# Force-clamp spectroscopy identifies a catch bond mechanism in a Gram-positive pathogen

Marion Mathelié-Guinlet [1,4], Felipe Viela [1,4], Giampiero Pietrocola [2], Pietro Speziale[2], David Alsteens [1,3✉] & Yves F. Dufrêne[1,3✉]

Physical forces have profound effects on cellular behavior, physiology, and disease. Perhaps the most intruiguing and fascinating example is the formation of catch-bonds that strengthen cellular adhesion under shear stresses. Today mannose-binding by the *Escherichia coli* FimH adhesin remains one of the rare microbial catch-bond thoroughly characterized at the molecular level. Here we provide a quantitative demonstration of a catch-bond in living Gram-positive pathogens using force-clamp spectroscopy. We show that the dock, lock, and latch interaction between staphylococcal surface protein SpsD and fibrinogen is strong, and exhibits an unusual catch-slip transition. The bond lifetime first grows with force, but ultimately decreases to behave as a slip bond beyond a critical force (~1 nN) that is orders of magnitude higher than for previously investigated complexes. This catch-bond, never reported for a staphylococcal adhesin, provides the pathogen with a mechanism to tightly control its adhesive function during colonization and infection.

[1] Louvain Institute of Biomolecular Science and Technology, UCLouvain, Croix du Sud, 4-5, bte L7.07.07, B-1348 Louvain-la-Neuve, Belgium. [2] Department of Molecular Medicine, Unit of Biochemistry, University of Pavia, Viale Taramelli 3/b, 27100 Pavia, Italy. [3] Walloon Excellence in Life sciences and Biotechnology (WELBIO), B-1300 Wavre, Belgium. [4]These authors contributed equally: Marion Mathelié-Guinlet, Felipe Viela. ✉email: david.alsteens@uclouvain.be; yves.dufrene@uclouvain.be

Mechanobiology is an exciting fast moving area aiming at understanding how physical forces influence cell behavior, physiology, and disease[1,2]. The vast majority of microbial cells attach to surfaces where they are subjected to mechanical stresses, such as hydrodynamic flow and cell–surface interactions. It has now become clear that such mechanical cues largely contribute to bacterial cell function and pathogenesis[3–5]. A prominent example is the formation of force-enhanced catch-bonds involving bacterial adhesins. The adhesion strength between a single adhesin and its ligand is defined as the force at which the complex will rupture, and is typically in the range of 50–250 pN[6,7] though much larger binding strenghts have been recently reported for several biomolecular complexes[8,9]. The lifetime of these specific bonds generally decreases with the applied force, a behavior known as slip bond. Counterintuitively, the lifetime of a catch bond grows with increasing tensile load up to a maximum and then decreases like a slip bond[10]. While catch-bonds have been suggested to potentially occur in various bacterial pathogens, today such a mechanism has only been identified and thoroughly chatacterized at the molecular level for the FimH adhesin from uropathogenic *Escherichia coli*[11–13]. Binding of FimH to mannose residues on epithelial cells is weak and short lived at low flow, whereas the bond strengthens at high flow, thus favoring pathogen adhesion during urinary tract infections. The underlying model is that mechanical force triggers an allosteric switch to a high-affinity, strong binding conformation of the adhesin.

During colonization of host tissues and biomaterials, staphylococci are exposed to extreme mechanical forces[5]. To withstand high physical stress, the pathogens have developed a variety of surface adhesins that tightly bind to host extracellular proteins such as the blood plasma protein fibrinogen (Fg)[14]. A variety of staphylococcal adhesins, including SdrG, ClfA, and ClfB, bind to Fg via the dock, lock, and latch (DLL) mechanism, whereby dynamic conformational changes of the adhesin result in a greatly stabilized adhesin–ligand complex. Following insertion of a short peptide sequence of Fg into a hydrophobic trench formed between the N2 and N3 subdomains of the adhesins, a conformational change at the C-terminus of N3 locks the peptide in place[15–17]. Single-molecule experiments have recently revealed that the DLL interaction is strong and enhanced under mechanical tension[18–22]. It has been speculated that this high mechanostability and force activation could result from a catch-bond mechanism, whereby the bound lifetime increases under force, meaning the DLL complex uses mechanical stress to strengthen. This notion is supported by flow chamber experiments showing that bacterial cell adhesion is enhanced by fluid shear stress[23,24].

While catch-binding represents an appealing explanation for DLL-based staphylococcal adhesion, a direct and unambiguous demonstration of such force-sensitive mechanism is still missing. Here we use force-clamp spectroscopy on living bacteria to demonstrate that staphylococcal adhesins are engaged in catch-bond interactions. We focus on the prototypical multifunctional adhesin SpsD from the pathogen *Staphylococcus pseudintermedius* ED99[25,26] (Fig. 1a), which is known to bind to Fg by a DLL interaction[26]. We show that the SpsD–Fg interaction is strong, and exhibits a catch-slip transition in response to force, a behavior never reported for a staphylococcal adhesin. Mechanical force first prolongs the lifetime of the bonds up to a critical force of 1100 pN, above which the bond lifetime decreases as an ordinary slip bond. Our results suggest that mechanical tension triggers structural changes in the adhesin so that the peptide ligand strongly binds in a shear geometry through long-lived hydrogen bonds. This study highlights the importance of protein mechanics in regulating the adhesion functions of bacterial pathogens.

## Results

**The interaction between SpsD and Fg is extremely strong.** The SpsD–Fg interaction was first studied by measuring the forces between single bacteria only expressing SpsD (thus not SpsL adhesins; thereafter, SpsD cells) and Fg-coated surfaces. Figure 1b and c show the adhesion forces and contour lengths obtained for three representative SpsD cells (for more cells, see Supplementary Fig. 1). Most adhesive events were strong with mean forces of 1873 ± 154 pN and contour lengths of 397 ± 81 nm (mean ± S.D.; $n = 207$ adhesive curves, 3 cells; for more retraction profiles, see Supplementary Fig. 3). These strong forces, averaged at 1812 ± 111 pN over 15 independent cells (Fig. 1d, Supplementary Fig. 4a), were specific to SpsD as they were abolished in ΔSpsD cells, mutants lacking both SpsD and SpsL adhesins (Fig. 1b inset). These mutant cells also showed a much lower adhesion probability as compared to SpsD cells (8 ± 6% vs 39 ± 19%, Fig. 1e). Mature SpsD adhesin is formed by 1031 residues (Fig. 1a) from which only 745 residues can be stretched upon pulling, as the N1 domain is not directly involved in the interaction and the SS region is not surface exposed. Considering that each amino acid contributes 0.36 nm to the contour length of the polypeptide chain[27], our contour lengths of ~400 nm suggest that both SpsD and Fg are being stretched upon pulling the cells away from the Fg-surfaces.

To study the mechanical strength of individual complexes, single adhesins were picked up and pulled with an AFM tip modified with Fg (Fig. 2). As in the above experiments, strong unbinding forces were measured between SpsD cells and Fg-tips (Fig. 2a; for more cells, see Supplementary Fig. 2), with a mean adhesion force of 1447 ± 157 pN and molecular contour length of 306 ± 83 nm (mean ± S.D.; $n = 1378$ adhesive curves; 3 cells; for more retraction profiles, see Supplementary Fig. 3). Again, these forces, averaged at 1518 ± 103 pN over 11 independent cells (Fig. 2d, Supplementary Fig. 4), were specific to SpsD as mutant cells were mostly exhibiting very low forces ~100 pN (Fig. 2d) and were almost non-adhesive (2 ± 1%) (Fig. 2e). The tiny differences observed in between cells and between single-cell and single-molecule experiments are likely due to variability of cells and random orientation of the Fg on the AFM tip. But, we cannot exclude that (i) the uncertainty on the calibration of the cantilever spring constant, and (ii) AFM datasets obtained with different cantilevers (thus of different spring constants, see Supplementary Tables 1 and 2)[28] also play a role in such small variability. We believe this strong interaction force originates from a DLL binding mechanism because: (i) biochemical analyses demonstrated that SpsD binds Fg through such mechanism, (ii) the high force, much stronger than that of typical protein–receptor interaction (from one to few hundreds of pN), is in the range of values recently reported for adhesins engaged in a DLL interaction[18,19]. While SpsD could bind more than one Fg molecule, we are confident that such high forces do not originate from the rupture of a variable number of weak bonds as similar sharp distributions were obtained in both single-cell and single-molecule experiments (Figs. 1 and 2). The weak SpsD–Fg bonds (<100 pN) recorded in both single-cell and single-molecule experiments could result either from a transit unspecific attachment of the bacterial surface or from a misoriented interaction. Indeed, Milles *et al.* have demonstrated for the DLL-formed SdrG–Fg complex that a lack of control in pulling geometry could result in a much weaker interaction: a strength of ~2500 pN was observed under physiologically relevant direction of force application as opposed to a strength of ~60 pN shown in a non-native tether geometry[19]. Finally, single-molecule mapping revealed that the SpsD strong binding events were localized in dense and packed areas on the bacterial cell surface, thus forming what we here call large clusters (Fig. 2c). DLL-forming adhesins

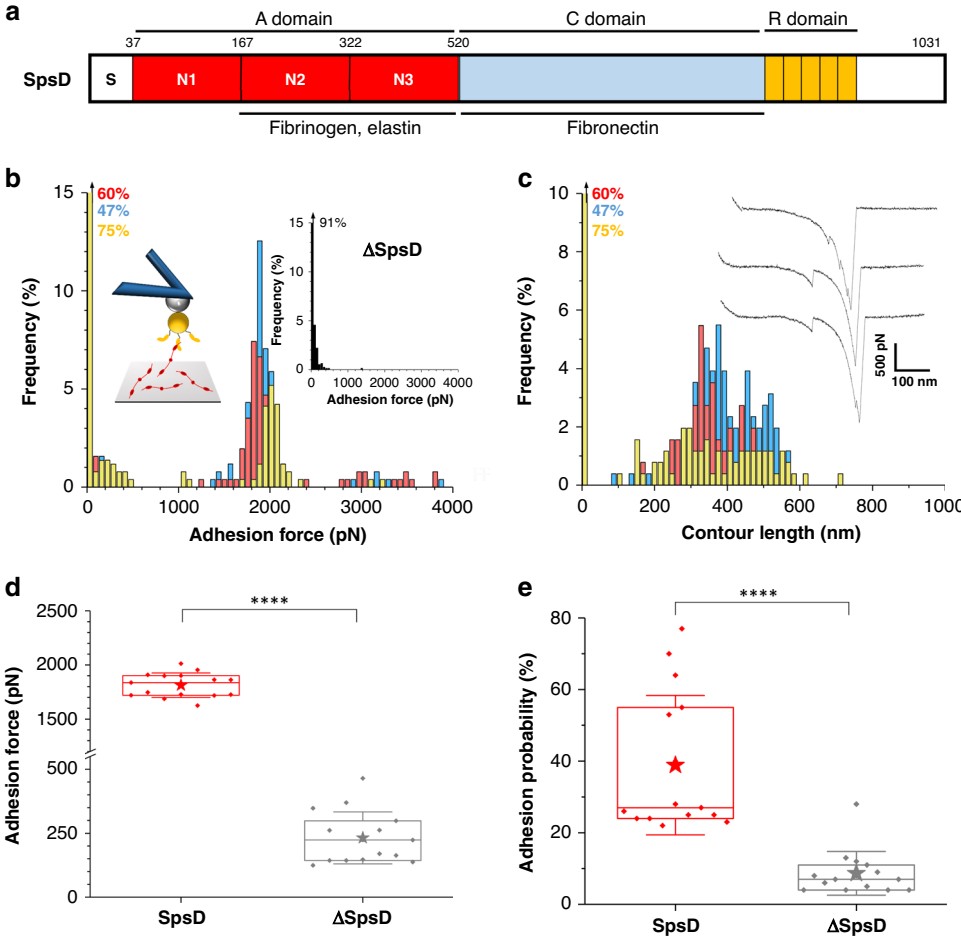

**Fig. 1 SpsD mediates strong bacterial adhesion to fibrinogen-surfaces.** (**a**) Schematic representation of the SpsD protein from *S. pseudintermedius* ED99. Following a signal sequence (S), SpsD exhibits an A domain, homologous to FnBPs and Clfs from *S. aureus*, a connecting region C and a repeat region R. Maximum adhesion force (**b**) and contour length (**c**) histograms obtained by recording force–distance curves in PBS between three different *S. pseudintermedius* ED99 SpsD cells and Fg-coated surfaces, at a retraction velocity of 1000 nm s$^{-1}$ (total number of curves from 3 cells $n = 207$). Insets in (**b**) show a scheme of the setup and the adhesion force histogram obtained on $\Delta$SpsD cells (3 cells merged). Inset in (**c**) shows representative retraction force profiles for SpsD cells. Box plots of the mean adhesion forces (**d**) and the mean adhesion probabilities (**e**) observed for 15 cells from 5 independent cultures. Mean adhesion forces were estimated from the sharp peaks (range of 1100–2300 pN) shown in (**c**). Stars are the mean values, lines the medians, boxes the 25–75% quartiles and whiskers the SD. Student *t*-test: ****$p \leq 0.0001$.

were shown to form similar clusters[18,22]. During colonization and invasion, adhesin clustering may promote the recruitment of host receptors such as integrins[29].

**Mechanical force enhances the SpsD–Fg interaction**. To investigate the dynamics of the interaction, we measured the binding force $F$ as a function of the loading rate ($LR$)—rate at which the mechanical force is applied—over 4 independent SpsD cells probed with retraction velocities ranging from 500 nm s$^{-1}$ to 10,000 nm s$^{-1}$ (see Supplementary Fig. 5). $LRs$ are extracted from the linear slope immediately preceding the rupture event on the force vs time curves (Fig. 3a inset). The dynamic force spectrum (DFS) shows dense data point clusters (due to different retraction velocities), sometimes overlapped, narrowly distributed around ~1500 pN (Fig. 3a). In the 1100–2300 pN force range, covering the sharp peaks observed in single-cell and single-molecule experiments (Figs. 1 and 2), the DFS distribution is well-fitted by the Bell Evans model[30], confirming the expected force loading-rate dependency of the rupture force. Using this model, we determined the following kinetic parameters: $x_\beta = 0.09 \pm 0.02$ nm and $k_{off} = 5 \times 10^{-13} \pm 4 \times 10^{-12}$ s$^{-1}$. Both parameters are comparable to the ones determined for SdrG binding to Fg through

the DLL ($x_\beta = 0.051$ nm, $k_{off} = 9.2 \times 10^{-11}$ s$^{-1}$)[19]. The very low values of $k_{off}$ obtained by means of AFM single-molecule experiments illustrate the high mechanostability of partners involved in a DLL interaction studied under external force. Interstingly, and as previously reported, this high mechanostability does not specifically correlate with the bulk binding affinity of SpsD for Fg found to be $0.360 \pm 0.032$ μM[26]. Finally, sorting adhesion forces by discrete ranges over $LR$ also demonstrated that the strong interaction is favored by tensile force (Fig. 3b).

**Force-clamp spectroscopy captures a catch-slip transition**. It has recently been speculated, yet never demonstrated, that the extreme strength of the DLL interaction might originate from a catch-bond mechanism[18–20,22,31]. While most biological bonds become weaker and dissociate faster when they are subjected to an external force, catch bonds resist rupture and become counter-intuitively longer-lived under force. To test whether this applied to SpsD, we performed single-molecule force clamp experiments (Fig. 4a; 11 bacteria from 6 independent cultures) in which the bond was clamped at loads increasing from 800 pN to 1300 pN, with a retraction velocity of 1000 nm s$^{-1}$ (see Supplementary Fig. 6). The bond lifetime ($\tau$) was first directly assessed from its

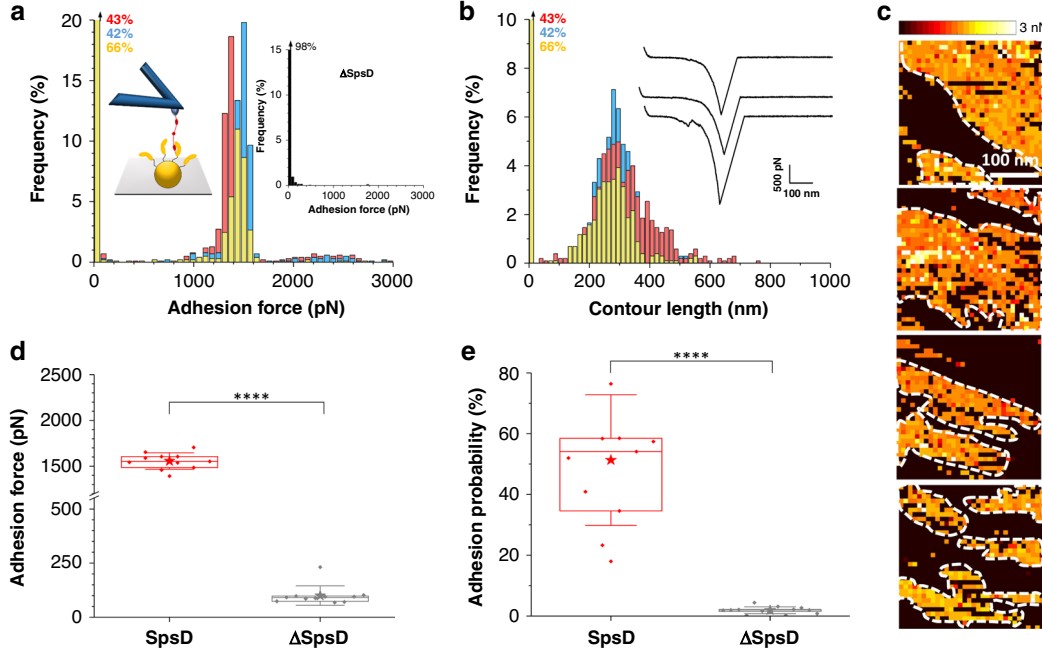

**Fig. 2 Imaging and functional analysis of single SpsD adhesins on living bacteria.** Maximum adhesion force (**a**) and contour length (**b**) histograms obtained by recording force–distance curves in PBS between three different *S. pseudintermedius* ED99 SpsD cells and AFM tips functionalized with Fg, at a retraction velocity of 1000 nm s$^{-1}$ (total number of curves from 3 cells $n = 1378$). Insets in (**a**) show a scheme of the setup and the adhesion force histogram obtained on ΔSpsD cells (3 cells merged). Inset in (**b**) shows representative retraction force profiles for SpsD cells. (**c**) Adhesion maps obtained on SpsD cells highlighting the clustering phenomenon (white dotted lines). (**d**, **e**) Box plots of the mean adhesion forces (**d**) and the mean adhesion probabilities (**e**) obtained for 11 cells from 5 independent cultures. Mean adhesion forces were estimated from the sharp peaks (range of 1100–2300 pN) shown in (**b**). Stars are the mean values, lines the medians, boxes the 25–75% quartiles and whiskers the SD. Student *t*-test: ****$p ≤ 0.0001$.

persistent time (Fig. 4a). For each clamping force, more than 100 spontaneous unbinding events were recorded. The extracted $\tau$ values showed a dispersion (Fig. 4b), which does not follow a Gaussian distribution, likely due to variability of cells and random orientation of the Fg on the AFM tip. Nonetheless, a non para-metric Kruskal Wallis test confimed the significant differences in lifetime reported for some pairs of loads. Mean lifetimes of 0.5, 0.8, 1.7, 2.0, 1.0, and 0.7 s were obtained for $F = 800$, 900, 1100, 1200, and 1300 pN respectively (Fig. 4d). Thus one clearly sees an increase in lifetime, from 0.5 s to 2.0 s, in the 800 to 1100 pN force range, while beyond this critical force, the lifetime switched back to lower lifetime values (<1.0 s). Below 1100 pN, the bond is longer-lived with increasing mechanical force (catch-bond behavior) while above this critical force, the bond weakens with further increasing force (slip-bond behavior). In addition, we have plotted the number of intact bonds vs time, i.e., the survival plot, to extract the bond lifetime[32] (Fig. 4c). Taking into account a time range that encompasses the highest density of data (0–2 s whatever the clamping force), the bond survival probabilities were mostly described by a single exponential decay. According to $N(t) = N_0 \exp(-t/\tau)$, the average lifetime $\tau$ could either be estimated from the linear slope $(-1/\tau)$ of the logarithmic repre-sentation of the survival plot or by the direct exponential fit of the data (see Supplementary Table 3). The $\tau$ values slightly differed from those obtained by averaging the lifetimes directly extracted from the raw data: 0.5, 0.6, 1.8, 2.3, 1.2, and 0.8 s were obtained for $F = 800$, 900, 1100, 1200, and 1300 pN, respectively (Fig. 4d). Yet, the catch-slip transition was still very clear with an even more pronounced transition at $F = 1100$ pN. We note that looking at the full time range an apparent transition at higher times can be observed for some clamping forces. None of the previous models (*e.g.*, multiexponential decay) could fit, even converge, on this full range (see Supplementary Fig. 7); a few

residual experimental data not reflecting the real mechanism could explain such behavior. Finally, we also showed that the rate of force application did not affect substantially the catch-slip bond transition (see Supplementary Fig. 8). Stretching the complex with a retraction velocity of 10,000 nm s$^{-1}$, instead of 1000 nm s$^{-1}$, did not alter the trend in lifetime values according to tensile force, and did not significantly impact the complex lifetime at the critical force threshold of ~1100 pN. This suggests that the clamp rate did not influence the interaction dynamics. Accordingly, these data provide a direct demonstration that the SpsD–Fg interaction follows a catch-slip bond transition, occur-ring at a well-defined critical force that is well beyond that of any other biological bonds.

## Discussion

Catch-binding represents an unusual and counterintuitive phe-nomenon of life. While stress-enhanced adhesion has been described for a variety of bacterial species, there is still very little direct, quantitative pieces of evidence for catch bond mechan-isms. This is partly due to the lack of advanced single-molecule methods capable to probe the mechanics of fully functional adhesins in living cells. We have identified and characterized a catch-bond mechanism in a Gram-positive pathogen. To our knowledge, single-molecule analysis of catch bonds had never been performed on living bacteria before. SpsD adhesins localize into clusters at the cell surface, which may enhance bacterial adhesion to host cells. Dynamic force spectroscopy shows that the SpsD-Fg DLL interaction is extremely strong and enhanced by mechanical tension. Most importantly, force-clamp spectroscopy reveals that SpsD and Fg form catch bonds that become longer lived in the presence of mechanical force. Beyond a critical force (1100 pN) the bond lifetime decreases and the complex behaves

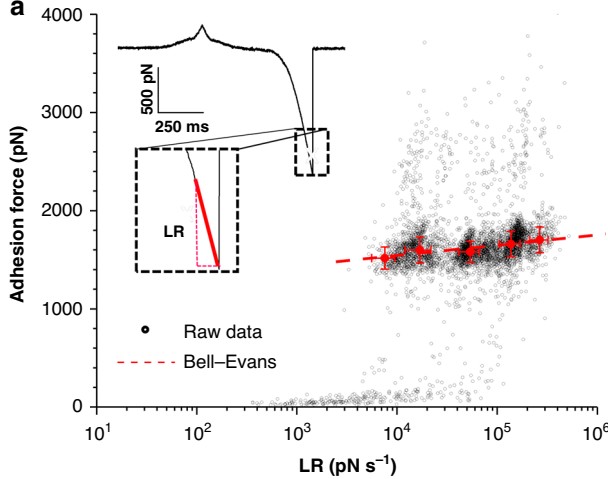

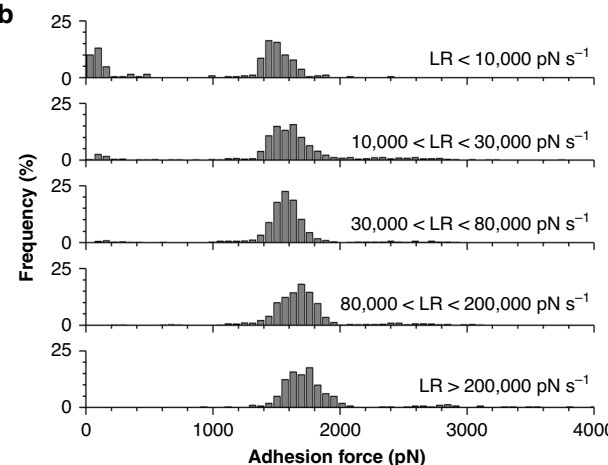

**Fig. 3 Binding of SpsD to fibrinogen is enhanced by tensile force.**
(**a**) Dynamic force spectroscopy plot obtained by recording force–distance curves in PBS between Fg-tips and *S. pseudintermedius* ED99 SpsD cells (*n* = 4326 data points from 4 cells). Inset shows a representative force vs time curve, highlighting the way the loading rate (*LR*) is extracted. A Bell Evans model fit (dashed red line) through the most-probable rupture force and force *LR* shows the expected force loading-rate dependency of the force ($x\beta$ = 0.090 nm $k_{off}$ = 5.07 × 10$^{-13}$ s$^{-1}$). (**b**) Corresponding rupture force histograms as a function of discrete ranges of *LRs*, emphasizing the shift toward higher forces with increasing *LRs*.

the proposed model states that under load the protein undergoes an allosteric switch from a low- to a high-affinity conformation. Other catch bond models have been proposed, including the sliding-rebinding, the one or two bound state—two pathways and the deformation models, all suggesting an energetic landscape in which the load applied acts as a force-dependant equilibration between potential different bound states and unbinding pathways[36–39]. In the later model, tensile force directly induces a structural change of the binding pocket resulting in a tighter fit. Further studies using e.g. simulations of crystal structure analysis are required to unravel the detailed mechanism of SpD catch binding. Nevertheless, it is tempting to speculate that it might involve force-induced changes in the adhesin binding pocket, in line with the well accepted notion that DLL binding involves dynamic conformational changes that lead to highly stable complexes and extreme mechanical strength. We note that current models describe catch binding under low tensile force, in the entropic regime where transition between states can occur, whereas the SpsD catch bound occurs in a very high-force regime, where the interacting molecules are subjected to extreme tension. We thus hypothesise that the SpsD catch bond may involve an unconventional model where propagation and direction of the applied force play central roles[8,31,40].

The formation of long-lived catch bonds may involve force-induced hydrogen bonds between the binding site and the peptide ligand. For the cadherin cell–cell adhesion protein, single-molecule force-clamp spectroscopy, molecular dynamics and steered molecular dynamics simulations have suggested that tensile force bends the cadherin extracellular region such that they form long-lived, force-induced hydrogen bonds that lock dimers into tighter contact[41]. Milles et al. showed that the extreme mechanical stability of DLL complexes formed by staphylococcal SdrG and ClfB adhesins originates from an intricate hydrogen bond network between the ligand peptide backbone and the adhesin[19]. The target peptide is confined in a screwlike manner in the binding pocket of SdrG and the binding strength of the complex results from numerous hydrogen bonds between the peptide backbone and the adhesin. These observations may suggest that the structural basis for the DLL catch-bond involves the force-induced formation of long-lived hydrogen bonds.

Finally, it is possible that this staphylococcal catch-bond has evolved in response to physical stress occurring during colonization, providing the pathogens with a mechanism to finely control their adhesive functions. Surface-attached staphylococci are exposed to mechanical cues associated with fluid flow, scraping, or epithelial turnover[42]. To cope with this, cell surface adhesins engage into various specific interactions, among which the strong, multistep DLL interaction. On the other hand, systemic dissemination is crucial for many bacterial pathogens, enabling them to spread from the initial site of infection to remote target tissues. One expects the catch bond phase (long-lived bonds) to strengthen colonization and biofilm formation under medium shear stress, whereas the slip phase (short-lived bonds) at extreme shear will help the bacteria to detach, disseminate and colonize new sites. SpsD is structurally and functionally related to DLL-forming adhesins from *S. aureus* including clinically relevant methicillin-resistant (MRSA) strains which cause nosocomial infections that are notoriously difficult to treat. This suggests that catch bonds involving force-induced hydrogen bonds might be generalized to all staphylococci, including MRSA ones.

as a slip bond. Flow chamber assays and single-molecule experiments have revealed the occurrence of catch-slip transitions in FimH, PSGL-1 ligand, actin/myosin, and T cell receptors with transition forces around 1–10 pN[11,32,33]. Truly unique to our study is the magnitude of the critical force, around two to three orders of magnitude higher than in all previously investigated complexes. These single-molecule results rationalize earlier observations showing that increased shear stress promotes staphylococcal adhesion[31,34,35], and that the DLL interaction (or variations of it) represents the strongest non-covalent biomolecular bond ever measured to date[18–20,22]. While the possibility of catch binding has been speculated in several recent reports, such catch-bond/slip-bond behavior was never demonstrated in a staphylococcal adhesin. We argue that identifying such a catch bond mechanisms among microbial pathogens represents a challenge in mechanobiology, especially as they might represent potential targets for new therapeutics.

What is the structural basis for the formation of the SpsD-Fg catch bond? In the widely investigated *E. coli* FimH adhesin[11,36],

## Methods
**Construction of *spsD*- and *spsL*-null bacterial mutants and growth conditions**. The single mutant strain lacking *spsL* gene, thus only expressing SpsD adhesins (here indicated as SpsD cell), and the double mutant lacking the genes *spsL* and

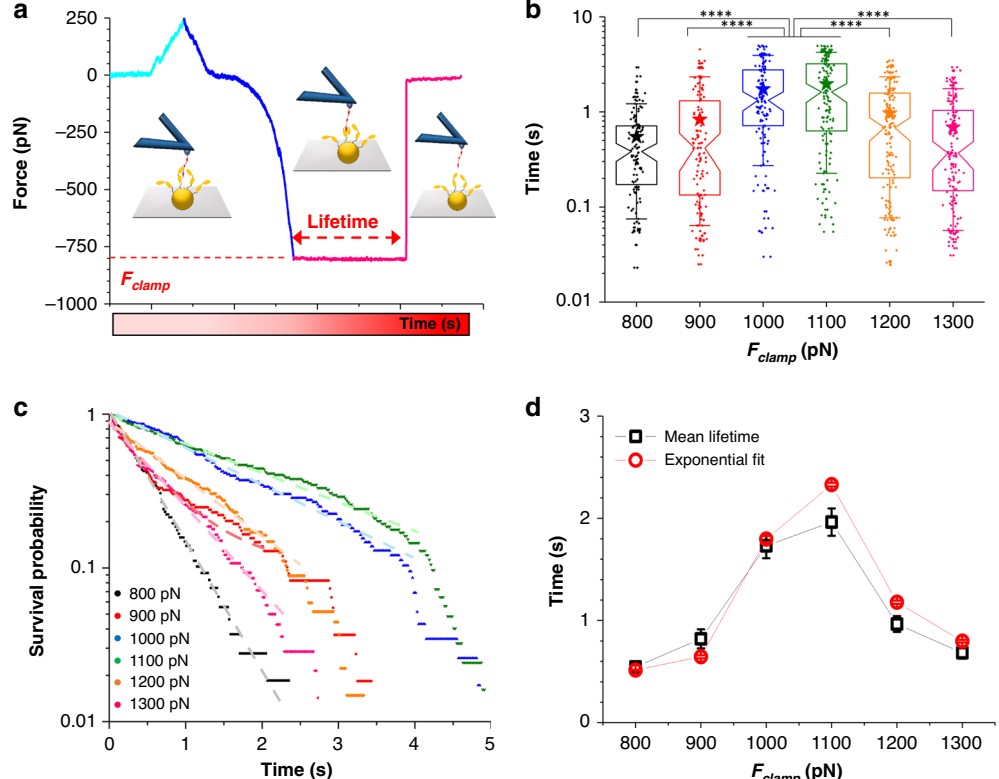

**Fig. 4 The DLL-interaction between SpsD and fibrinogen involves a catch-slip bond transition.** (**a**) Characteristic force–clamp curve (force vs time) along with illustrations of the setup at different steps of the clamp process. Cyan stands for the approach, blue for the retraction, and pink for the clamp and spontaneous rupture of the SpsD-fibrinogen bond. The bond lifetimes is given by the constant force regime in red. (**b**) Box plot (overlapped with data, $n = 108$, 109, 116, 124, 135, and 140 for $F_{clamp} = 800$, 900, 1000, 1100, 1200, 1300 pN, respectively, recorded on 11 cells from 6 independent cultures) of the lifetimes extracted from the force vs time curves at a retraction velocity of 1000 nm s$^{-1}$. Stars are the mean values, boxes the 25–75% quartiles and whiskers the 10–90% interval. Kruskal–Wallis test followed by Dunn's multiple-comparison test: ****$p \leq 0.0001$. (**c**) Bond survival probabilities for the SpsD–Fg interactions, measured at six different clamping forces. Fits are presented as dashed lines with corresponding colors. (**d**) Force-dependent bond lifetimes, extracted from the exponential fit on the survival plot and the average lifetimes from the box plot, at a retraction velocity of 1000 nm s$^{-1}$, exhibit a catch-slip transition. For average lifetimes, error bars represent the standard deviation from $n = 108$, 109, 116, 124, 135, and 140 data points for $F_{clamp} = 800$, 900, 1000, 1100, 1200, 1300 pN, respectively, on 11 cells from 6 independent cultures (as in (**b**)). For lifetimes extracted from the exponential fit, error bars stand for the standard error of the fit.

spsD (here referred to as ΔSpsD cell) were generated from *S. pseudintermedius* ED99, a strain isolated from a canine bacterial pyoderma[43]. The complete genome sequence of *S. pseudintermedius* ED99 was deposited in the Gene Bank database under the accession number CP002478[44]. Sequence of SpsD is available at link https://www.ncbi.nlm.nih.gov/protein/ADX76659.1/, accession number ADX76659.1. All strains were grown in brain heart infusion (BHI) broth overnight, at 37 °C and under shaking at 200 rpm, to reach their stationary phase. For AFM experiments, cells were harvested by centrifugation at 3000 g for 5 min and washed twice with PBS.

**Functionalization of substrates and cantilevers with Fg.** Human Fg was obtained from Calbiochem (Merck, Darmstadt, Germany) and further purified on a gelatin-Sepharose column to remove contaminating fibronectin. Gold-coated glass coverslips and gold cantilevers (OMCL-TR400PB-1, Olympus Ltd., Tokyo, Japan) were immersed overnight in an ethanol solution containing 1 mM of 10% 16-mercaptododecahexanoic acid/90% 1-mercapto-1-undecanol (Sigma), rinsed with ethanol and dried with N2. Substrates and cantilevers were then immersed for 30 min into a solution containing 10 mg mL$^{-1}$ N-hydroxysuccinimide (NHS) and 25 mg mL$^{-1}$ 1-ethyl-3-(3- dimethylaminopropyl)-carbodiimide (EDC) (Sigma) and rinsed with ultrapure water. Finally they were incubated with 0.1 mg mL$^{-1}$ of Fg for 1 h, rinsed further with PBS buffer, and then immediately used without de-wetting.

**Single-cell force spectroscopy.** Colloidal probes were prepared as described earlier[45]. Briefly, a small amount of UV-curable glue is spread on one side of a cover glass while on the other side silica microspheres are deposited. Under the control of a NanoWizard IV atomic force microscope (JPK Instruments) coupled to an optical microscope, a triangular-shaped tip-less cantilever is brought manually into contact with a small droplet of glue and then approach to catch an

isolated single silica bead. The newly designed colloidal probe is finally put under a UV lamp for 15 min. to cure the glue. Before any experiment, the colloidal probe is put 1 h in a 10 mM Tris–HCl buffer solution (pH 8.5) containing 4 mg mL$^{-1}$ dopamine hydrochloride, and washed in the same buffer. The nominal spring constant of such cantilevers was determined by the thermal noise method, giving an average value of ~0.08 N m$^{-1}$. For single-cell experiments, 50 μL of a suspension of ca. 1 × 10$^6$ cells were transferred into a glass petri dish containing Fg-coated substrates on the other corner, the whole being immersed in PBS. The colloidal probe was brought into contact with a bacterium, which is first caught through electrostatic interactions with polydopamine. The cell probe was then positioned over Fg-substrates without de-wetting. Cell probes were used to measure interaction forces on Fg-surfaces at room temperature by recording multiple force curves (16 × 16) on different spots with a maximum applied force of 250 pN, and approach and retraction speeds of 1000 nm s$^{-1}$.

**Single-molecule force spectroscopy.** For SMFS experiments, gold cantilevers ($k \sim 0.02$ N m$^{-1}$) were prepared as described above and bacteria were immobilized on polystyrene substrates. SMFS measurements were performed at room temperature in PBS buffer with a NanoWizard IV AFM. Adhesion maps were obtained by recording 32 × 32 force-distance curves on areas of 500 nm × 500 nm with an applied force of 250 pN, a constant approach and retraction speed of 1000 nm s$^{-1}$. In total 11 cells of each strain from 5 independent cultures were probed. For loading rate experiments, arrays of 32 by 32, force curves were recorded on 500 nm × 500 nm areas at increasing retraction speeds: 0.5, 1, 3, 5, and 10 μm s$^{-1}$. A total of 4 cells from 2 independent cultures were probed.

**Single-cell and single-molecule data analysis.** Data were analyzed with the data processing software from JPK Instruments (Berlin, Germany). Considering the high forces (~1500 pN) involved in the SpsD–Fg complex, an extensible worm-like

chain (WLC) model, taking into account the sretching of the backbone chain reached at such forces[46], was used to extract the rupture force and contour length of the last specific peak in each curve, according to the following equation[46]:

$$F(x) = \frac{k_B T}{l_p} \left[ \frac{1}{4} \left( 1 - \frac{x}{L} + \frac{F}{\varphi} \right)^{-2} + \frac{x}{L} - \frac{1}{4} \right] \quad (1)$$

where $\varphi$ is the stiffness and $l_p$ the persistence length of the molecule, both free parameters in the model. Distribution of $F$ and $L$ were then plotted and further analyzed with Origin.

**Force clamp experiments**. For force clamp experiments, gold cantilevers ($k \sim 0.06$ N m$^{-1}$) were prepared as described above and bacteria were immobilized on polystyrene substrates. Fg-modified tips were used to map ($10 \times 10$ pixels) 300 nm × 300 nm areas on top of the bacteria. First, the cantilever is brought into contact with the bacterial surface for a dwell time of 100 ms, at a velocity of 1000 nm s$^{-1}$ and with an applied force of 250 pN. Second, the cantilever is moved apart from the sample at a constant speed (either 1000 nm s$^{-1}$ or 10,000 nm s$^{-1}$) until a trigger force in the range [800; 1300 pN]. Finally, the ligand-receptor bond is clamped at the trigger force thanks to a controlled feedback loop that adjusts continuously the cantilever-sample distance to maintain the force constant. The persistent time of the bond gave the lifetime of the molecular complex at the clamping force. Bond survival probabilities were calculated by distributing the raw lifetimes obtained in each pixel of the map into time bins (interpolated histograms), and as a function of the clamping force. Briefly, for each clamping force, all raw data are gathered and sorted as histograms (the bin size did not influence the survivals). The cumulative frequencies ($fc$) obtained are then translated into survival ($S$) data: $S = (100 - fc)/100$. Mean complex lifetimes were then estimated (i) either by fitting the slope of the logarithmic decay obtained on survival plots (or similarly direct exponential fit on the data) or (ii) by averaging the raw lifetimes extracted from the maps. Given the regularization times of the PID loops for force clamp feedback (~10 ms), we only took into account lifetime data larger than 20 ms.

**Statistical methods**. The statistical significance of differences among bacterial strains adhesions and bond lifetimes was assessed using GraphPad Prism 8. Student $t$-test was used when the distribution was normal; otherwise the Kruskal–Wallis test was favored followed by Dunn's multiple-comparison test. $P$ values when differences are significant are provided on graphs and in figure legends.

## Data availability
Data supporting the findings of this manuscript are available from the corresponding authors upon reasonable request. A reporting summary for this article is available as a Supplementary Information file. Source data are provided with this paper.

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

## Acknowledgments

Work at the Université catholique de Louvain was supported by the European Research Council (ERC) under the European Union's Horizon 2020 research and innovation programme (grant agreement no. 693630), the FNRS-WELBIO (grant no. WELBIO-CR-2015A-05 to Y.D.F. and grant no. WELBIO-2019S-01 to D.A.), the National Fund for Scientific Research (FNRS), and the Research Department of the Communauté française de Belgique (Concerted Research Action). Funding by the Fondazione CARIPLO (Grant Vaccines 2009-3546) to P.S. is acknowledged. D.A. and Y.F.D. are Research Associate and Research Director at the FNRS. We thank Ross Fitzgerald for providing mutant strains.

## Author contributions

M.M.G., F.V., G.P., P.S., D.A., and Y.F.D. designed the experiments, analyzed the data, and wrote the article. M.M.G. and F.V. collected the data.

## Competing interests

The authors declare no competing interests.
