## [Peer Review File · Nature Communications]

REVIEWER COMMENTS

Reviewer #1 (Remarks to the Author):

Overall, I think this paper made significant efforts using two different experimental models of investigating the interaction between SpsD and Fg in a single cell and single molecule level and reported that the SpsD-Fg interaction was first, studied by measuring the forces between single bacteria and Fg-coated surfaces. The observed catch bond in SpsD was important. It would be great to confirm the deformation model associated with the DLL steps: docking of the Fg peptide in a pocket formed by the N2-N3 subdomains of SpsD, the subsequent conformational change of the N3 subdomain that extends its C-terminus over the N2 subdomain, which finally locks the Fg in a perfect shear geometry through an intricate hydrogen bond network using crystallography or molecular simulation studies. Such studies can also be helpful to further confirm the structural basis for the formation of SpsD-Fg catch bonds. Your study interestingly extends the observations catch bond protein mechanics which regulated in the adhesion functions of bacterial pathogens.

But I advise for major revision with the following specific comments. I strongly suggest that the authors drop over-the-top claims which are mainly rooted in assumptions are not sufficiently supported by the data. Otherwise, if the authors want to maintain these strong claims, then more experiments are needed to support these claims. Also, the data representation could be further improved as well.

- 1, To address a fully extended SpsD protein length and the assumption that SpsD and, to some extent Fg, are being stretched upon pulling the cells away from the Fg-surfaces, the authors can assess the crystal structural information

2. In general, the number of experimental methods, apart from the low repeat number, is limited. The majority of observations were left to assumption, utilizing further experiments such as a co-crystal structure (mentioned above), investigating the possibility of rupture in the protein bounds similar to the separation of the FimHL and FimHp. and perhaps molecular dynamics simulations experiments are required or structural analysis provided for discussion.

Minor comments:

1. line 79: The bacteria interact with more than one protein on the cell face is not unlikely. I think this needs to be mentioned.

2. Line 81 & Fig1 (Also Fig 2): Discussing the mean value indicates that the population is considered as normally distributed. Then the far left and right data on the histogram are outliers, and I would merge the data and use a scatter plot or boxplot to display the range. I will also merge the figure B and C, as figure C is the control for panel B.

3. Line 89: I believe a transit attachment to the surface or a weak and misoriented attachments, could also be reasons for this observation.

4. Line 94: Given this one of the main findings of the paper, I would like to see alternative experimental suggestion to re-evaluate this point or explain why unspecific binding is not a likely explanation for this observation.

5. Fig2b: Looking at figure 2b, panels I am not sure, which force value (from 0-3pN) has been selected as representative of single molecules. Assuming the 3pN value was the baseline, then only on the second panel from the top there might be two clusters. Apart from this, the area covered by non-zero pixels is quite different between the panels. I would mark the clusters if my comments are incorrect. Otherwise please explain more to make this clear and add a scatter plot of the clusters next to the pixel values.

6. Line 107 & 108, please report the number of independent experiments.

7. Fig 3b: I will also plot Adhesion force as a function of LR, using a line plot with error bars.

8. Line 116: misspelled "continuously"

9. Line 140: The differed in values from the survival plots, are likely to be due to variability in protein expression among cells, but the catch-slip transition still required more repeats to be reported as a clear observation.

10. Line 145: misspelled "occurring"
11. Line 151: misspelled "Positive"
12. Line 155: misspelled "by"
13. Line 157: misspelled "importantly"
14. Line 157 also, a strong comment for a limited number of experiments. To fully confirm, these following experiments are potentially required; a co-crystal structure, possibility of rapture in the protein bounds similar to the separation of the FimHL and FimHp. and perhaps molecular dynamics simulations.
15. Line 165: misspelled "occurrence"
16. Line 202: misspelled "occurring"
17. Line 212: This suggests that catch-bonds involving force-induced hydrogen bonds required potential experimental methods suggestion to further confirm the involvement of hydrogen bonds in the catch-bonds between staphylococcal surface protein SpsD and fibrinogen. Also, an additional explanation of why this is (or is not) important, is required.
18. Line 344, fig 3: the data is from one experiment 3 independent cells or in 3 experiments?
19. Line 357 fig4: These cells are from how many independent experiments?
20. Extended Data Figure 1, 2, and 3: Please include the number of independent experiments conducted.

Reviewer #2 (Remarks to the Author):

The present manuscript describes catch bonding behavior in a gram-positive pathogen adhering to human hosts targets. It is the first report of this kind of interaction in the context of gram-positive bacteria. Topic and relevance of results match Nature Communications very well. The described catch bonding behavior is an exciting result that is interesting to a broad audience, especially in the context of pathogen adhesion.

At this stage I cannot recommend publication of this manuscript. While the core of the data, i.e. the high interaction forces are a sound result, a number of misconceptions, and methodological fallacies in the analysis and subsequent interpretation must be corrected. Most of these can be resolved through study of established literature and use of appropriate models.

Major revisions in analysis are required and stronger direct evidence for catch-bonding must be provided. The methods given in this version are not complete, and not sufficient to potentially replicate the work (e.g. sequences of constructs, crucial information on which strains were used is missing, exact measurement conditions are incomplete etc.). The supplementary materials should be greatly expanded with details. Mere references to previous work for methods are not acceptable, also per the guidelines of Nature Communications.

I have confidence that the authors will be able to correct these shortcomings quickly and directly, to strengthen the manuscript for publication in Nature Communications.

These non-exhaustive following points should be addressed directly – if necessary with additional experiments. They are given in no particular order:

The heart of this work is the force clamp spectroscopy showing a short bond lifetime at lower (800 pN) and high (1300 pN) forces with a bond lifetime peak at around 1100 pN. While compelling, a more rigorous data analysis scheme is required. It is also concerning that the crucial data points at constant forces of 800 and 900 pN, that show shorter lifetimes than higher forces they key to catch bond behavior, have the lowest statistics (a mere 11 curves for 900 pN)

The distributions of the data from 900 to 1100 pN seem hardly different and the mean values provided are dominated by outliers, what would the values look like if these were excluded? The key data here is the data set at around 800 pN, as it shows the short lifetimes required to establish the inverted U-shape of lifetimes for a catch bond model. A number of questions arise for this central finding in Fig.

4:

Especially the values for lifetimes reported of less than 10 ms should be shown explicitly as force vs time plots as in Fig 4b. How were such short lifetimes be determined without extreme uncertainty? Especially in Supplementary Figure 3a at 800 pN these make out 5 out of 18 datapoints. The PID loops for force clamp feedback have regularization times on the order of more than 10 ms, so that these values are accurate measurements seems very unlikely. Indeed, Figure 4a clearly shows oscillations of the instrument in the force clamp traces, which are in the order of tens of milliseconds, making sub 10 ms lifetimes measured hard to accept as accurate – or even valid. What are typical feedback times for the instruments employed here and what is its detection limit? I would be inclined to dismiss any lifetime data smaller than 20 ms out of an abundance of caution given feedback times and the corresponding artefacts in Fig. 4a. This should not affect the catch bond results if average lifetimes at 800 pN are on the order of 500 ms

Overall, a more rigorous analysis of the lifetime distributions is required, with statistical tests that show the significance between the lifetime differences reported, e.g. simple KS-test. The current data is too scattered and low in number to be fully convincing with high significance. Especially, a direct comparison between Fig 4e and Supplementary Figure 3c raises questions, e.g. why do the mean lifetimes in Supp. Fig 3c plateau between 1000 and 1200 pN but not in Fig 4e?

The fits underlying the determined exponential fit tau values are not shown, but should be prominently included in a main figure, also what are the uncertainties of the fits? These fits provide a much more pronounced inverted U-shape of bond lifetimes, so they should be displayed and discussed explicitly including an analysis of their uncertainties.

In Figure 4d) what is shown here exactly, is this an interpolated histogram for different forces, how was this plot assembled?

Also, in Fig. 4e) why are the uncertainty in rupture force so drastically different for different force values, especially the vanishing uncertainty at 900 pN?

If the catch bond claim, which seems very plausible for this system and has been proposed a number of times, is to hold a more rigorous analysis with clear-cut significance and statistics for the force dependent lifetimes must be given. I am confident that the authors can provide this analysis.

The force activation proposed here rests on a dynamic force spectrum, and a correlation of force loading rate and rupture force. This correlation is an unavoidable consequence of the measurement mode of constant velocity with polymer linkers, but here falsely attributed to the specific properties of the complex investigated here:

Line 108 ff.

“Sorting adhesion forces by discrete ranges over LR also demonstrated that the strong interaction is enhanced by tensile force (Fig. 3b).”

This is a tautological statement: For a constant velocity AFM pulling experiment such as this, at a constant retraction velocity a higher rupture force will always imply a higher force loading rate at that rupture force. This is due to the entropically elastic linker which enforces the changing slope of the force-extension curve. At higher forces the slope of the force extension and thus force-time curve is steeper than at lower forces, which automatically results in a higher force loading rate. This is very evident in the force extension (extension linearly coupled to time via the constant retraction velocity) curves in Figure 1, their slope increases monotonically.

Or shorter: given the entropically elastic linkers used here, as e.g. described by the WLC model $F=F(x, l_p, l_c)$, the force loading rate \dot{F} is a function of rupture force F , $\dot{F}(F)$ or given extension and using the WLC model $\dot{F}(F(x, l_p, l_c))$, given fixed contour length l_c , and persistence length l_p).

To correct for this effect a force ramp experiment with a controlled, constant force loading rate could be conducted. A good discussion, including a corrected Bell-Evans model, for this effect can be found here: Ray, Brown, Akhremitchev 2007 “Correction of Systematic Errors in Single-Molecule Force

Spectroscopy with Polymeric Tethers by Atomic Force Microscopy." Journal of Physical Chemistry B
Thus, any interpretation, most notably the loading rate dependent force activation, arising from this phenomenon should be redacted from the manuscript.
Furthermore, the authors must specify how exactly the force loading rate was determined, it is missing from the methods.

In similar arguments the K_m given here as a "spring constant of the complex" is another proxy of the entropic linker elasticity in this system (essentially being a similar fit as the force loading rate discussed above, merely with time replace by distance x , which is linearly related to time by the constant retraction velocity).

Again, at lower forces the slope enforced by the here more pronounced entropic elasticity of the protein polymer will lead to flatter slopes of Force versus distance, only to increase at higher forces when extension approaches the tethered polymers contour length. The entire discussion of k_m being an intrinsic property of the molecular complex is invalid. K_m fitted here is rather the effective stiffness of the cantilever + entropically elastic linker system (by the authors' own calculation on the order of around 1000 amino acids or hundreds of nm in contour length). This inherently increases at higher forces. In other words, measured here (again how was K_m fit exactly, please provide methods) is not the molecular complex, but the dominating part is the entropic elasticity of the linker and unfolded polypeptide chain that connect the folded adhesin to its ligand.

It is thus no surprise that the K_m measured at high forces, where the force extension relation is almost linear and the effect of the linkers entropic elasticity becomes smaller as it is almost fully extended, is on the same order of magnitude (around 0.04 N/m) as the spring constant of the AFM cantilever (around 0.08 N/m).

Furthermore, an almost linear force extension behavior shows that the polymer has reached an enthalpic regime where backbone bond stretching becomes an issue, a general issue at forces much larger than 500 pN. The authors should consider (optionally) fitting an appropriate polymer elasticity model to their data, such as Livadaru, Netz, and Kreuzer, Stretching Response of Discrete Semiflexible Polymers, *Macromolecules*, 2003 with QM corrections for backbone stretching given in (Hugel et al., Highly Stretched Single Polymers: Atomic-Force-Microscope Experiments Versus Ab-Initio Theory, *PRL* 2005) which become relevant at these extremely high forces and would give them quantitative contour and persistence length values for their force-extension curves.

In Fig. 3a it is unclear which retraction velocities gave which data point cluster, these should be color coded to show which retraction velocity was used. The large spread of high force rupture forces if not acquired from a single cantilever may be cantilever calibration artifacts and should be discussed.

Lastly, the rupture forces around 60 pN with low force loading rates are most likely non-specific interactions. Given that a live bacterium's surface is not blank, a discussion of how specific from non-specific events were discerned should be provided. Especially as the target molecule fibrinogen is rapidly coating surfaces and very sticky.

Low (~ 100 pN) force unbinding events appear in the negative control in Fig 1. Yet, they are discussed as specific "weak adhesion force" and evidence for force activation in the dynamic force spectra. Likely, these low-force events are unspecific attachment of bacterium to surface, or fibrinogen. Please resolve this unclarity with reference to the control experiments.

Secondly the attachment strategy used to anchor the Fg target by NHS/EDC chemistries is not site-specific meaning multiple pulling geometries of Fg are possible (e.g. from N- or C-terminus, or even both termini). Previous work cited has shown a decisive influence of pulling geometry in rupture force for SpsD's homolog SdrG (60 pN non-natively, over 2000 natively) The possibility of such non-native low force geometries occurring here should be discussed, these could also be the cause of low force interactions.

The statement that Bell-Evans or Friddle models do not apply here is not substantiated by the data shown. Indeed, the authors do not even try to fit these models here. A Bell-Evans fit to the rupture

force histogram at high forces e.g. in Fig. 3b or Supplementray Figures 1 and 2, should be performed, judging by the shape of the distribution it should converge and yield a zero force off-rate and a distance to the transition state that could also be used to compare to such parameters acquired for six other DLL adhesins referenced in previous work.

Have the authors considered fitting explicit catch bond models directly to their data, such the sliding-rebinding model as in Rakshit et al. PNAS 2012 for cadherins?

The discussion part proposes that the DLL mechanism only completes, or rather deforms to form a catch bond, upon force application. However, previous crystallographic and thermodynamic studies for these adhesins have shown that the bound state of a target inside the binding cleft between N2 and N3 domain, and a closed "latch" is achieved upon, and even required for, target binding, most notably: Bowden et al., Evidence for the "dock, lock, and latch" ligand binding mechanism of the staphylococcal microbial surface component recognizing adhesive matrix molecules (MSCRAMM) SdrG, JBC 2008

Thus, it seems unlikely, all the while possible, that mechanical forces enhance this binding process, definitive evidence for this hypothesis would require to show that force application enhances the complexes' kinetic on-rate.

The aspect of force causing major conformational changes to create a catch-bond is worth of debate, indeed the work referenced for the formation of backbone hydrogens bonds showed no discernible or large conformational changes in the adhesin:ligand complex in MD simulations for SdrG – which clashes with the proposed deformation model. However, these results only apply to SdrG, SpsD may be a homologous, albeit mechanistically very different adhesin.

The methods are incomplete. Constructs and strains used should be given in exact sequence, Vectors (if used) and strains used should be specified exactly with databank accession numbers, protein sequences, plasmids etc. used should be given in great detail to enable reproduction of this work. Currently, it is unclear which sequences were used here.

Unfolding peaks preceding receptor ligand unbinding in curves are never discussed, the sequence of the gene would not suggest that these be present. To these arise out of the stretched Fg, or additional interactions? The peaks preceding complex rupture in curves as seen in curves in Fig 1b should be analyzed for repeated contour length increments that would indicate a specific domain unfolding and discussed.

Further comments in order of appearance in the manuscript:

Abstract:

The abstract claims that the only microbial catch bond mechanism discussed in molecular detail is FimH. Nord et al. PNAS 2017 have dissected catch bonding in rotor/stator interactions of the bacterial flagellar motor.

The wording of "holy grail", both stylistically and factually seems out of place here. Especially, since catch-bonding mechanisms have already been resolved in pathogens such as in the case of FimH as cited by the authors.

Please specify how catch bonding mechanisms will lead to antibacterial strategies.

Line 46 ff.

Adhesion ligand systems such as Titin:Telethonin, Streptavidin:Biotin, or the Cohesin Dockerin Type III system and its variants with similar functions are far above these values, 400 to over 1000 pN. (Bertz et al., PNAS 2009; Sedlak et al., Sci. Adv. 2020; Bernardi et al. JACS 2019). 250 pN seems a low value for typical interactions especially considering more recent work.

Line 55:

Please expand and reference the occurrence of these “extreme mechanical forces”

Results:

Line 84 ff.

“that its folded length is ~20 nm”

How do the authors arrive at that distance, please provide a reference or a detailed explanation? How would the 1031 amino acids of the sequence result in just 20 nm of effective folded length? The value of 1031 amino acids used to calculate the contour length is incorrect. Firstly, the N1 domain is not involved in ligand binding so it’s around 130 amino acids do not count into the contour length. Secondly, raw sequence of the SpsD gene shown in Fig. 1a is also processed during export and sortase-mediated attachment, which should be discussed as it will change the contour length of the protein - it will likely be shorter in its mature form. Please recalculate these values.

The authors mention the length of the Fg used, but would also need to justify where it was attached and how that contributes to its expected contour length.

Please provide a reference for the contour length per amino acid.

In Figure 1 the rupture distance but not the contour length is given, please fit appropriate models to the force extension curves to arrive at a contour length, or at least provide a rationalization or estimate of the contour length of the stretched complex.

Fig.1 what retraction velocity was used to acquire the data shown?

Line 95:

Please provide references and details for how a DLL mechanism was shown for this adhesin. High forces alone are not an argument for the DLL mechanism, it is conceivable – yet unlikely given SpsD’s high homology to established adhesins like SdrG - that this is a different or at least varied binding mechanism.

Please provide an explanation for the unbinding force differences of around 400 pN between Fig. 1 and Fig. 2 for essentially the same system unbinding. Could cantilever calibration errors play a role here?

The authors argue that clustering of adhesins on the bacterial surface show that single complexes unbind in their data. This argument should be refined as receptor clusters make binding events of multiple adhesins leading to higher forces more likely. The rupture force populations at around 3000 pN could be the source of this cluster binding, please discuss.

Line 148:

With several catch bonds reported (even directionally asymmetric ones such as by Huang et al. Science 2017 for vinculin/actin) or Notch-Jagged (Luca et al. Science 2016) I would hardly characterize them as a “controversial” phenomenon.

Please provide statistics for the experiments, curve attempts, successful single interactions, measurement time, and show more raw force extension curves and force clamp curves in the supplement.

Cantilever spring constants may influence the system under investigation. Please provide the exact spring constants measured from cantilevers associated with each measurement in the supplement. Please provide a detailed calibration method including InvOLS determination.

Reviewer #3 (Remarks to the Author):

Referee Report: "Force-clamp spectroscopy identifies a catch-bond mechanism in a Gram-positive pathogen," by Mathelié-Guinlet et al.

In their paper "Force-clamp spectroscopy identifies a catch-bond mechanism in a Gram-positive pathogen," Mathelié-Guinlet et al. investigate the dock, lock, and latch (DLL) interaction between staphylococcal surface protein SpsD and fibrinogen using atomic force microscopy in the dynamic force spectroscopy mode and in force-clamp spectroscopy mode. In dynamic force spectroscopy, authors find unusually high bond rupture forces (1 – 2 nN) above force loading rates of 1,000 pN/s and much lower rupture forces (approx. 60 pN). Using force clamp spectroscopy, the authors show, that forces up to 1,100 pN enhance the bond strength and increase the mean bond lifetime (up to approx. 6 s), while at forces above 1,100 pN the bond weakens and the bond lifetime decreases again. Owing in large extent to the use of the force clamp method (Figure 4), this paper represents indeed the first study which unambiguously and convincingly demonstrates catch-bond behavior for adhesion molecules of Gram-positive bacteria. The study is well done and the paper well written. The study represents a major breakthrough in mechanobiology and provides a better understanding of infection mechanisms of Gram-positive bacteria. The paper should therefore clearly be published by Nature Communications, after the authors, have addressed the following (minor) shortcomings of their manuscript:

1. As mentioned above, owing to the force clamp method used, this is the first paper clearly and unambiguously demonstrating catch-bond behavior in adhesion molecules from Gram-positive bacteria. Nevertheless, it is not the first AFM study pointing to catch bond behavior in such adhesins. The authors should make this clear in the introduction and discussion and reference previous studies which observed catch-bond like characteristics in other receptor ligand systems of Gram-positive bacteria. An overview can be found in reference 3 of this manuscript as well as in ACS Nano, 13, 7155-7165 (DOI:10.1021/acsnano.9b02587)

2. Fig. 4d and line 130: "The bond survival probabilities at almost each force clamp were described by a single exponential decay."

The curves at 1000 pN (blue) and 1100 pN (green) clamp force in Fig. 4d clearly show multi-exponential decay. The other curves also show deviations from a single exponential behavior. The authors should include the fits of $N(t)$ which were used to determine τ in Fig. 4d. This would help the reader to judge whether the decay is single or rather multi-exponential. As mentioned, the curves at 1000 pN (blue) and 1100 pN (green) clamp force show multi-exponential decay. A bi-exponential kinetic model might be more suitable here. Examples of such models can be found e.g. in ACS Nano, 6, 1314-1321, (DOI: 10.1021/nn204111w) and in Angewandte Chemie Int. Ed., 58 (29), 9787-9790 (DOI: 10.1002/anie.201902752)

Please show fits, and consider a multi-exponential fit at 1000 pN and 1100 pN. The authors should also change Fig. 4e accordingly, by giving both lifetimes, when a bi-exponential kinetic model fits the data better than a single exponential model, and discuss possible reasons for the bi-exponential kinetics in the paper.

3. Lines 114 – 120 and Fig. 3C. The author report differences in the molecular spring constant (k_m) in the different force regimes and conclude that "This strongly supports a deformation model in which tensile force induces a switch from a soft, weak binding conformation, to a stiff, strong-binding conformation, resulting in a mechanically stabilized adhesin-ligand complex."

The molecular spring constant proteins and polypeptides is highly non-linear and increase in molecular spring constant with force is a normal behavior of these molecules. This holds true for any protein. Thus, this observation does not explain or support catch-bond behavior. The relevant parameter here is not the molecular spring constant, but the stretch modulus (S) which can be obtained by fitting an extensible worm like chain model to the force extension curves, as it is commonly done in the field. The authors should show S rather than k_m , or remove lines 114-120 and Fig. 3C from the manuscript.

4. Fig. 2 and Fig. 3: The authors show data from only 3 cells in these figures and move the data from three more cells to the Extended data. I would recommend to merge data from Fig. 2 and 3 and Extended Data and show data from all six cells in the main figures (2 and 3).

5. Lines 85/86: „Assuming that (i) the processed mature SpsD adhesin comprises 1,031 residues, (ii) that its folded length is ~ 20 nm“
Please give a reference

6. Line 88: “This suggests that 87 SpsD and, to some extent Fg, are being stretched upon pulling the cells away from the Fg-surfaces.” If ruptures occur at 351 ± 62 nm, why is Fn also stretched? Do you assume that SpsD remains partially folded at these forces?

7. Line 155: Typo: my=> by

8. Fig. 4a/line 353: the black and blue curve are difficult to discriminate in the printed manuscript. Consider using another color.

9. Line 355: Typo: ... curves obtained at different loads, and...

10. Line 414: How were the bacteria immobilized on PS?

General comments for the editor and for the referees

We thank the referees and the editor for their very relevant and encouraging comments. We feel that our revised manuscript strongly improved owing to the numerous suggestions made by the referees and additional experiments performed over the past months as well as a more rigorous data analysis. Specifically, we:

- increased the number of cells in each set of classical single-cell ($n = 15$ cells from 5 independent cultures for both SpsD and mutant strains) and single molecule experiments ($n = 11$ cells from 5 independent cultures for both SpsD and mutant strains) (see new figures 1, 2 and supplementary figures 1 to 4)
- extended the statistics for all clamping forces in force clamp experiments both at low ($v = 1 \mu\text{m/s}$) and high ($v = 10 \mu\text{m/s}$) loading rates with 11 cells probes among 6 independent cultures ($n > 100$ events per clamping force), (see new figure 4 and supplementary figures 5 and 6)
- re-analysed, in light of the referee comments, the single-cell and single-molecule data with an extended worm like chain model, considering the stretching of the backbone chain occurring at $\sim nN$ forces (see new figures 1, 2 and supplementary figures 1 to 4)
- checked the relevance of the classical Bell-Evans model on the DFS data (see new figure 3)
- tested the classical models, which actually could not match and explain the observed catch bond (see new discussion)

Overall, referee 1 suggested additional experiments using crystallography and / or molecular simulations. Though very interesting, such experiments are highly time-consuming and would require initiation of new collaborations with top-experts as completely different background and equipments are needed. We feel such works represent new stories on their own, and are thus beyond the scope of the present work. Moreover, we would like to highlight that our single-molecule analysis on live cells is the most direct and quantitative way to demonstrate a catch bond. To our knowledge, single-molecule analysis of catch bonds had never been performed on living bacteria before, meaning our study is outstanding from both the microbiology and nanophysics standpoints.

In summary, owing to our new experiments and analysis we identify the first medically-relevant catch-bond mechanism in a staphylococcal adhesin, with the bond exhibiting the typical transition from a catch bond (long-lived bond) to slip bond (short-lived bond) with an extremely high threshold force of **$\sim 1100 \text{ pN}$** . As discussed in our manuscript, earlier experiments have revealed the occurrence of catch-slip transitions in FimH, PSGL-1 ligand, actin/myosin, and T cell receptors with transition forces around **$1-10 \text{ pN}$** . So truly unique to our study is the magnitude of the critical force, around two to three orders of magnitude higher than in all previously investigated complexes.

Reviewer #1 (Remarks to the Author):

Overall, I think this paper made significant efforts using two different experimental models of investigating the interaction between SpsD and Fg in a single cell and single molecule level and reported that the SpsD-Fg interaction was first, studied by measuring the forces between single bacteria and Fg-coated surfaces. The observed catch bond in SpsD was important.

We thank the reviewer for the encouraging and positive feedback. Below, we have answered point-by-point all questions, hoping the revised manuscript now clarifies all the concerns.

It would be great to confirm the deformation model associated with the DLL steps: docking of the Fg peptide in a pocket formed by the N2-N3 subdomains of SpsD, the subsequent conformational change of the N3 subdomain that extends its C-terminus over the N2 subdomain, which finally locks the Fg in a perfect shear geometry through an intricate hydrogen bond network using crystallography or molecular simulation studies. Such studied can also be helpful to further confirm the structural basis for the formation of SpsD-Fg catch bonds. Your study interestingly extends the observations catch bond protein mechanics which regulated in the adhesion functions of bacterial pathogens.

Pietrocola *et al.* already showed that SpsD binds Fg through the DLL mechanism or a variant of it (PLoS ONE 8, e66901 (2013)). The authors constructed Δ LL and trench mutants of the N2N3 subdomains of the A domain of SpsD and showed that binding to the γ -chain of Fg was completely abrogated in these cases, demonstrating the interaction between SpsD and Fg occurs by DLL. The aim of the present work is to investigate the strength and dynamics of this interaction. We found that the interaction is extremely strong, similar to a covalent bond. Most importantly we demonstrate, on living bacteria and at the single-molecule level, that it follows a catch bond behavior, which had never been shown for a staphylococcal adhesin. We would like to stress that our single-molecule analysis on live cells is the most direct and quantitative way to demonstrate a catch bond. This allowed us to unambiguously identify the first medically-relevant catch-bond mechanism in a staphylococcal adhesin, with the bond exhibiting the typical transition from a catch bond (long-lived bond) to slip bond (short-lived bond) with an extremely high threshold force of ~ 1100 pN (see new discussion). As discussed in our manuscript, earlier experiments have revealed the occurrence of catch-slip transitions in FimH, PSGL-1 ligand, actin/myosin, and T cell receptors with transition forces around 1-10 pN. So truly unique to our study is the magnitude of the critical force, around two to three orders of magnitude higher than in all previously investigated complexes. While crystallography and molecular simulations would be very interesting, such experiments are highly time-consuming and would require initiation of new collaborations with top-experts with different background and equipments are needed. These analyses represent new stories on their own, and are thus beyond the scope of the present work.

But I advise for major revision with the following specific comments. I strongly suggest that the authors drop over-the-top claims which are mainly rooted in assumptions are not sufficiently supported by the data. Otherwise, if the authors want to maintain these strong claims, then more experiments are needed to support these claims. Also, the data representation could be further improved as well.

We believe that the direct demonstration of a catch bond is strongly supported by our data (see above and manuscript). Again, we stress that our AFM force clamp experiments, under

physiological conditions, and on living bacteria, are the most direct and quantitative demonstration for catch bonds (compared to data obtained on purified adhesins). However, we agree that the discussion of the underlying molecular mechanism remains speculative and we have therefore qualified the text on lines 185-212. We want to highlight that, in our SpsD-Fg system, none of the known theoretical models were adequately fitting the experimental lifetimes observed, likely due to the extreme tension under which the molecules are subjected. Consequently, we hypothesize that the SpsD catch bond may involve an unconventional model where propagation and direction of the applied force play central roles (see *e.g.* Nano Lett. 15, 7370–7376 (2015), J. Am. Chem. Soc. 141, 14752–14763 (2019)).

1, To address a fully extended SpsD protein length and the assumption that SpsD and, to some extent Fg, are being stretched upon pulling the cells away from the Fg-surfaces, the authors can assess the crystal structural information.

While both adhesin and Fg are likely to be stretched, the extent to which is unclear. Therefore we have qualified the discussion on lines 85-91.

2. In general, the number of experimental methods, apart from the low repeat number, is limited. The majority of observations were left to assumption, utilizing further experiments such as a co-crystal structure (mentioned above), investigating the possibility of rupture in the protein bounds similar to the separation of the FimHL and FimHp. and perhaps molecular dynamics simulations experiments are required or structural analysis provided for discussion.

We would like to stress that unlike traditional bioassays where multiple methods are combined, we are using here a range of advanced, state of the art AFM techniques (thus not only one), which are only developed in a few labs worldwide: single-cell force spectroscopy (Fig. 1), single-molecule force spectroscopy (Figs. 2, 3), and most importantly force clamp (Fig. 4), which to our knowledge has never been demonstrated on living bacteria like here.

As far as the low repeat number is concerned, we have greatly improved the statistics by performing much more experiments:

- n > 10 cells from at 5 independent cultures (both SpsD and mutant) in single-cell and single molecule experiments (Figs. 2 and 3)
- n > 100 events from 11 cells among 6 independent cultures per clamping force (Fig. 4)

Owing to these advanced experiments, we identify the first medically-relevant catch-bond mechanism in a staphylococcal adhesin, with the bond exhibiting the typical transition from a catch bond (long-lived bond) to slip bond (short-lived bond) with an extremely high threshold force of ~1,100 pN. Truly unique to our study is the magnitude of the critical force, around two to three orders of magnitude higher than in all previously investigated complexes.

Minor comments:

1. line 79: The bacteria interact with more than one protein on the cell face is not unlikely. I think this needs to be mentioned.

Given the similar sharp distributions of forces obtained in both single-cell (Fig. 1) and single-molecule (Fig. 2), we are confident that the interactions mostly represent single-molecules. In single-molecule analysis (Fig. 2) we have developed protocols over the years which have been published in details that ensure single-molecule detection. However, it is indeed not unlikely that SpsD could bind to more than one Fg molecule. We have included a comment on lines 105-106.

2. Line 81 & Fig1 (Also Fig 2): Discussing the mean value indicates that the population is considered as normally distributed. Then the far left and right data on the histogram are outliers, and I would merge the data and use a scatter plot or boxplot to display the range. I will also merge the figure B and C, as figure C is the control for panel B.

We agree with the reviewer that the far left and right data can be considered as outliers in the previous distributions shown in figures 1 and 2. This is pretty obvious when looking at the sharp peak observed in the nN range in both cases. Indeed, we only account for these data in the range of 1,100 – 2,300 pN, seen as normally distributed, to calculate our mean values. The way we calculated our mean values is now included in the caption of corresponding figures.

In the initial manuscript, we have indeed presented the control experiments in a different panel in figures 1 and 2, to also show a scheme of the setup and raw force curves. Following the advice of the reviewer we have changed figures 1 and 2 in the revised manuscript, to include the control experiments as insets. As required by the reviewer, we have also added adhesion force boxplots and adhesion probabilities for all the cells in main figures 1 and 2 and supplementary figure 4.

3. Line 89: I believe a transit attachment to the surface or a weak and misoriented attachments, could also be reasons for this observation.

We apologize for our claims related to the weak interaction. Previous work has notably shown that the lack of control on Fg geometry on the AFM tip could lead to weak interactions (50 - 60 pN) with ClfB, also known to engage with Fg through a variation of the DLL mechanism (Science 359, 1527–1533 (2018)). Following the reviewer's advice, we have now reformulated our claims considering the weak interactions as a potential artefact of pulling geometry on lines 108-113.

4. Line 94: Given this one of the main findings of the paper, I would like to see alternative experimental suggestion to re-evaluate this point or explain why unspecific binding is not a likely explanation for this observation.

As mentioned above, we have now reformulated our strong claims concerning the weak interactions and interpretations derived from it on lines 185-212.

5. Fig2b: Looking at figure 2b, panels I am not sure, which force value (from 0-3pN) has been selected as representative of single molecules. Assuming the 3pN value was the baseline, then only on the second panel from the top there might be two clusters. Apart from this, the area covered by non-zero pixels is quite different between the panels. I would mark the clusters if my comments are incorrect. Otherwise please explain more to make this clear and add a scatter plot of the clusters next to the pixel values.

We apologize if the claim on nanoclusters was misleading for the reviewer. To make it clearer, we have now indicated the borders of the clusters in each map of figure 2b. We

define a cluster as a region of the cell surface (probed by the AFM tip) where specific strong interactions between SpsD and Fg concentrate, namely regions of bright pixels. As can be seen in figure 2b, the bright pixels are rather packed in dense regions, reminiscent of clusters. We have also mentioned this definition in lines 114-117.

6. Line 107 & 108, please report the number of independent experiments.

In the revised manuscript, the number of cells and independent experiments are now clarified for each set of data, in the main text, in the Methods section and in the captions of each figure.

7. Fig 3b: I will also plot Adhesion force as a function of LR, using a line plot with error bars.

In new figure 3, the high forces in the nN range were sorted according to discrete ranges of LR and considered for Bell Evans fitting (see red dots and red dotted line). In this regime, the data are well fitted by the Bell Evans model, confirming the force loading rate dependency of the rupture force.

8. Line 116: misspelled “continuously”

This typo has been corrected.

9. Line 140: The differed in values from the survival plots, are likely to be due to variability in protein expression among cells, but the catch-slip transition still required more repeats to be reported as a clear observation.

As mentioned above, more experiments have been performed to extend our statistics on force clamp results. They are presented in new figure 4: at least 100 data, collected on 11 cells from 6 independent cultures, have been recorded for each clamping force. We have also performed more experiments to improve the statistics at a retraction velocity of 10 $\mu\text{m/s}$, reaching similar statistics than those with $v = 1 \mu\text{m/s}$ (see new Supplementary Fig.6).

10. Line 145: misspelled “occurring”

11. Line 151: misspelled “Positive”

12. Line 155: misspelled “by”

13. Line 157: misspelled “importantly”

These 4 typos have been corrected.

14. Line 157 also, a strong comment for a limited number of experiments. To fully confirm, these following experiments are potentially required; a co-crystal structure, possibility of rapture in the protein bounds similar to the separation of the FimHL and FimHp. and perhaps molecular dynamics simulations.

As described above, we have improved our data increasing the statistics for force clamp experiments and including statistical significance tests (see new figure 4 and Supplementary Fig. 6). As mentioned before force clamp experiments are the most straightforward and quantitative technique to described catch bonds.

15. Line 165: misspelled “occurrence”

16. Line 202: misspelled “occurring”

These 2 typos have been corrected.

17. Line 212: This suggests that catch-bonds involving force-induced hydrogen bonds required potential experimental methods suggestion to further confirm the involvement of hydrogen bonds in the catch-bonds between staphylococcal surface protein SpsD and fibrinogen. Also, an additional explanation of why this is (or is not) important, is required.

In an elegant recent Science paper by Milles *et al.* (Science 359, 1527–1533 (2018)), single molecule experiments and molecular dynamics, demonstrated that the extreme mechanostability of DLL interaction, SdrG but also other DLL adhesins like ClfB arises from the formation of an H bond network between the adhesin and the ligand. They suggested that such phenomenon might result from a catch bond behavior, but without any proof, which is precisely what we demonstrate here. Considering this unambiguous study and that SpsD is a DLL adhesin we feel it is completely fair to suggest that our catch bond involves the formation of such H bond network, as explained in lines 202-212.

18. Line 344, fig 3: the data is from one experiment 3 independent cells or in 3 experiments?

19. Line 357 fig4: These cells are from how many independent experiments?

We have added now in the figure caption the number of independent experiments.

20. Extended Data Figure 1, 2, and 3: Please include the number of independent experiments conducted.

For comments 18 to 20, the number of independent experiments and cells have been clarified along the revised manuscript.

Reviewer #2 (Remarks to the Author):

The present manuscript describes catch bonding behavior in a gram-positive pathogen adhering to human hosts targets. It is the first report of this kind of interaction in the context of gram-positive bacteria. Topic and relevance of results match Nature Communications very well. The described catch bonding behavior is an exciting result that is interesting to a broad audience, especially in the context of pathogen adhesion. At this stage I cannot recommend publication of this manuscript. While the core of the data, i.e. the high interaction forces are a sound result, a number of misconceptions, and methodological fallacies in the analysis and subsequent interpretation must be corrected. Most of these can be resolved through study of established literature and use of appropriate models. Major revisions in analysis are required and stronger direct evidence for catch-bonding must be provided.

We thank the reviewer for the valuable and encouraging comments concerning the relevance and importance of our manuscript. These comments helped us to strengthen our manuscript. Below, we have answered point-by-point all questions. See also above our general comment. Especially, we have strongly improved the statistics obtained on force clamp experiments and also reconsidered our analysis of single-cell and single molecule data (see below).

The methods given in this version are not complete, and not sufficient to potentially replicate the work (e.g. sequences of constructs, crucial information on which strains were used is missing, exact measurement conditions are incomplete etc.). The supplementary materials should be greatly expanded with details. Mere references to previous work for methods are not acceptable, also per the guidelines of Nature Communications. I have confidence that the authors will be able to correct these shortcomings quickly and directly, to strengthen the manuscript for publication in Nature Communications.

We apologize for this lack of explanation in the Methods section, which has now been extended to provide any detail required to replicate the work. Concerning the sequence of constructs, it has already been published by our collaborators Pietrocola *et al.* (*Infect. Immun.* **83**, 4093–4102 (2015)). A brief overview of the strains is now provided in the Methods section lines 228-231. Furthermore, we have extended the Supplementary Information to provide more examples of histograms, raw curves and all results obtained per cell.

These non-exhaustive following points should be addressed directly – if necessary with additional experiments. They are given in no particular order:

The heart of this work is the force clamp spectroscopy showing a short bond lifetime at lower (800 pN) and high (1300 pN) forces with a bond lifetime peak at around 1100 pN. While compelling, a more rigorous data analysis scheme is required. It is also concerning that the crucial data points at constant forces of 800 and 900 pN, that show shorter lifetimes than higher forces they key to catch bond behavior, have the lowest statistics (a mere 11 curves for 900 pN) The distributions of the data from 900 to 1100 pN seem hardly different and the mean values provided are dominated by outliers, what would the values look like if these were excluded?

We agree with the reviewer that the distributions provided in the initial manuscript might have been misleading due to low statistics and outliers. To support more firmly our catch bond behavior, we have extended the statistics obtained for all forces, especially those at lower forces (800 and 900 pN). The revised manuscript now includes $n = 108, 109, 116, 124, 135$ and 140 events for $F = 800, 900, 1,000, 1,100, 1,200, 1,300$ pN respectively, recorded on 11 independent cells from 6 independent cultures. While doing so, we still observe a clear catch – slip transition with maximum lifetime at a force around 1,100 pN. However, as can be seen in new figure 4, the distribution is quite broad for each clamping force, which is likely due to the variability of cells and tip functionalization protocols in which Fg orientation cannot be properly controlled (see line 139). Nonetheless, given our non-normal distributions, we have performed a non-parametric statistical test (a Kruskal Wallis test) to compare all clamping forces by pair (see new figure 4 and Supplementary Fig. 6). This has confirmed the significant differences between lifetimes obtained at 800, 900, 1200 and 1300 pN with those obtained at 1000 and 1100 pN with $p < 0.0001$, demonstrating the catch-slip transition at those later forces for which no significant differences were reported.

The key data here is the data set at around 800 pN, as it shows the short lifetimes required to establish the inverted U-shape of lifetimes for a catch bond model. A number of questions arise for this central finding in Fig. 4:

Especially the values for lifetimes reported of less than 10 ms should be shown explicitly as force vs time plots as in Fig 4b. How were such short lifetimes be determined without extreme uncertainty? Especially in Supplementary Figure 3a at 800 pN these make out 5 out of 18 datapoints. The PID loops for force clamp feedback have regularization times on the order of more than 10 ms, so that these values are accurate measurements seems very unlikely. Indeed, Figure 4a clearly shows oscillations of the instrument in the force clamp traces, which are in the order of tens of milliseconds, making sub 10 ms lifetimes measured hard to accept as accurate – or even valid. What are typical feedback times for the instruments employed here and what is its detection limit? I would be inclined to dismiss any lifetime data smaller than 20 ms out of an abundance of caution given feedback times and the corresponding artefacts in Fig. 4a. This should not affect the catch bond results if average lifetimes at 800 pN are on the order of 500 ms

We agree with the reviewer and we did not consider any value below 20 ms in the revised manuscript. As mentioned by the reviewer, those (very few) data did not influence the outcome of our force clamp experiments and interpretations. This information has been added to the Methods section in lines 290-292.

Overall, a more rigorous analysis of the lifetime distributions is required, with statistical tests that show the significance between the lifetime differences reported, e.g. simple KS-test. The current data is too scattered and low in number to be fully convincing with high significance. Especially, a direct comparison between Fig 4e and Supplementary Figure 3c raises questions, e.g. why do the mean lifetimes in Supp. Fig 3c plateau between 1000 and 1200 pN but not in Fig 4e?

See above concerning the statistics and significant differences reported.

We agree with the reviewer that initial Figure 4e and Supp. Figure 3c differ in the complex lifetime reached at the critical force thresholds of 1,000-1,100 pN. But, the trend with a first increase in lifetime and then a decrease is still there, yet less pronounced. These data are obtained at two different retraction speed before clamping the SpsD-Fg interaction.

However, after performing significance tests on lifetime data, we do not see a significant difference at the two different retraction speeds at the forces of the catch bond, as mentioned on lines 156-158.

The fits underlying the determined exponential fit tau values are not shown, but should be prominently included in a main figure, also what are the uncertainties of the fits? These fits provide a much more pronounced inverted U-shape of bond lifetimes, so they should be displayed and discussed explicitly including an analysis of their uncertainties. The fits are now displayed in new figure 4 and supplementary figure 6. We also include in the table 1 below the goodness of the fit. After increasing the statistics for the force clamp experiments, analyzing and fitting the data, the lifetime values, extracted from the raw data (box plots) and fits (survival), are very close and the inverted U-shaped tendency is pretty similar (new figure 4d).

We want to clarify that the survival plots are built from the cumulative frequencies obtained on raw data (see below). Consequently, we feel that the uncertainties obtained on the survival fits are not representative of the real error we could get from our experiments. Indeed, in this case, each data point of the survival plot depends on the previous one, so that uncertainty values for the fitting parameters are extremely small (see table 1).

Table 1. Fitting parameters obtained by fitting the survival plots at different clamping force by a single exponential decay.

	Lifetime (s) (mean ± SE)	χ^2	R^2
800 pN	0,514 ± 0,003	$1,2 \cdot 10^{-4}$	0,998
900 pN	0,645 ± 0,015	$1,1 \cdot 10^{-3}$	0,969
1000 pN	1,797 ± 0,031	$1,1 \cdot 10^{-3}$	0,975
1100 pN	2,330 ± 0,017	$1,3 \cdot 10^{-4}$	0,994
1200 pN	1,179 ± 0,016	$5,7 \cdot 10^{-4}$	0,986
1300 pN	0,797 ± 0,013	$8,1 \cdot 10^{-4}$	0,983

However, we also provide, for each clamping force, the boxplots of all the lifetime values directly extracted from the rupture event in F vs time curves. There, one can observed the real distributions of the bond lifetimes and still one can clearly see the same inverted U-shaped tendency. With those box plots, that are not usually presented in papers, one can have the real / true dispersion of the bond lifetime. And still, the differences between the lifetime at {800; 900; 1200; 1300 pN} and {1000; 1100 pN} are significantly different ($p < 0,0001$).

In Figure 4d) what is shown here exactly, is this an interpolated histogram for different forces, how was this plot assembled?

The survival plot is indeed an interpolated histogram for different forces. Briefly, for each clamping force, all raw data are gathered (as presented with box plots in figure x) and sorted as histograms: between 0 and 5 seconds with a bin size of 0.02 ms (the bin size does not impact the final result). The cumulative frequencies (f_c) obtained are then translated into survival (S) data:

$$S = \frac{100 - f_c}{100}$$

These details are now included in the Methods section lines 284-288.

Also, in Fig. 4e) why are the uncertainty in rupture force so drastically different for different force values, especially the vanishing uncertainty at 900 pN?

We apologize if these uncertainties on force were misleading. Actually, in the initial figure 4e, we reported the “maximum adhesion force” reached after retracting the cantilever at the different setpoints and before clamping the force at those setpoints (i.e. clamping force). This explains the different uncertainties for the different forces, which were not the clamping forces. We acknowledge that this “maximum adhesion” was not the right parameter to plot there, and we have now reported the true clamping forces (as in the initial figures 4c-d for instance) which are actually almost not subjected to errors (similar to noise).

If the catch bond claim, which seems very plausible for this system and has been proposed a number of times, is to hold a more rigorous analysis with clear-cut significance and statistics for the force dependent lifetimes must be given. I am confident that the authors can provide this analysis.

Catch bonds in staphylococci have been suggested in a few earlier studies without any direct molecular proof for this. By contrast, we believe our single-molecule analysis on live cells presented here is the most direct and quantitative way to demonstrate a catch bond, meaning our conclusion is strong rather than "plausible". Having said that, we fully agree there was a need for improved statistics, significance and fitting, please see details above.

The force activation proposed here rests on a dynamic force spectrum, and a correlation of force loading rate and rupture force. This correlation is an unavoidable consequence of the measurement mode of constant velocity with polymer linkers, but here falsely attributed to the specific properties of the complex investigated here:

Line 108 ff.

“Sorting adhesion forces by discrete ranges over LR also demonstrated that the strong interaction is enhanced by tensile force (Fig. 3b).”

This is a tautological statement: For a constant velocity AFM pulling experiment such as this, at a constant retraction velocity a higher rupture force will always imply a higher force loading rate at that rupture force. This is due to the entropically elastic linker which enforces the changing slope of the force-extension curve. At higher forces the slope of the force extension and thus force-time curve is steeper than at lower forces, which automatically results in a higher force loading rate. This is very evident in the force extension (extension linearly coupled to time via the constant retraction velocity) curves in Figure 1, their slope increases monotonically. Or shorter: given the entropically elastic linkers used here, as e.g. described by the WLC model $F=F(x, l_p, L_c)$, the force loading rate \dot{F} is a function of rupture force F , $\dot{F}(F)$ or given extension and using the WLC model $\dot{F}(F(x, l_p, L_c))$, given fixed contour length L_c , and persistence length l_p).

To correct for this effect a force ramp experiment with a controlled, constant force loading rate could be conducted. A good discussion, including a corrected Bell-Evans model, for this effect can be found here: Ray, Brown, Akhremitchev 2007 “Correction of Systematic Errors in Single-Molecule Force Spectroscopy with Polymeric Tethers by Atomic Force Microscopy.” Journal of Physical Chemistry B

Thus, any interpretation, most notably the loading rate dependent force activation, arising from this phenomenon should be redacted from the manuscript.

Furthermore, the authors must specify how exactly the force loading rate was determined, it is missing from the methods.

First, let's specify how the loading rates (LR) were extracted from our data. For each force curve showing a specific adhesion event, we have plotted the corresponding force vs time curve and estimated the apparent LR from the linear slope just preceding the unbinding event (see lines 121-122 and new Fig. 3a). Consequently, the LR is not considered the same for all data taken at the same retraction velocity, as some papers do ($LR = v * k_c$, with k_c being the cantilever spring constant). Moreover, we analyzed the time interval during which the slope of the F vs time is constant. This time interval is shown in figure 1 below. We observed that the time interval minimum is ≈ 40 ms. This strongly suggests that, in the regime preceding the unbinding, our LR can be considered as constant, assumptions that are at the basis of the Bell-Evans and Friddle-Noy models unlike more complex models (e.g. Dudko-Hummer-Szabo model).

Figure 1. Distribution of the time interval over which the slope of the curve is linear (over $N = 70$ curves recorded at a retraction velocity of $1,000 \text{ pN}\cdot\text{s}^{-1}$).

Regarding the weak interaction and the mechanical activation of the bond, we have removed such strong claim from the manuscript.

Third, after performing more experiments and reanalyzing the data, we have now considered only the high forces $\sim 1.5 \text{ nN}$ for fitting by appropriate models. A linear increase of force with LR can be observed in the new figure 3a, well fitted by the Bell Evans model. Still, as the reviewer is pointing, the Bell Evans model assumes a constant loading rate, which is not true in case of stretching a polymeric tether, especially at these high forces $> 1 \text{ nN}$. Indeed, when force is applied to the molecular SpsD-Fg complex through the flexible adhesin, the loading rate is not constant anymore due to elastic stretching of the complex, which, in turn, modify the energetic properties of the bound. However, as we said above, we are extracting the LR from the very last part of the curve before the rupture, where the slope is linear (see also new figure 3a inset). Consequently, we feel that our approach is fair and sufficient in the context of the present paper.

In similar arguments the K_m given here as a “spring constant of the complex” is another proxy of the entropic linker elasticity in this system (essentially being a similar fit as the force loading rate discussed above, merely with time replace by distance x , which is linearly related to time by the constant retraction velocity).

Again, at lower forces the slope enforced by the here more pronounced entropic elasticity of the protein polymer will lead to flatter slopes of Force versus distance, only to increase at higher forces when extension approaches the tethered polymers contour length. The entire discussion of k_m being an intrinsic property of the molecular complex is invalid. k_m fitted here is rather the effective stiffness of the cantilever + entropically elastic linker system (by the authors' own calculation on the order of around 1000 amino acids or hundreds of nm in contour length). This inherently increases at higher forces. In other words, measured here (again how was k_m fit exactly, please provide methods) is not the molecular complex, but the dominating part is the entropic elasticity of the linker and unfolded polypeptide chain that connect the folded adhesin to its ligand.

It is thus no surprise that the k_m measured at high forces, where the force extension relation is almost linear and the effect of the linkers entropic elasticity becomes smaller as it is almost fully extended, is on the same order of magnitude (around 0.04 N/m) as the spring constant of the AFM cantilever (around 0.08 N/m).

Furthermore, an almost linear force extension behavior shows that the polymer has reached an enthalpic regime where backbone bond stretching becomes an issue, a general issue at forces much larger than 500 pN. The authors should consider (optionally) fitting an appropriate polymer elasticity model to their data, such as Livadaru, Netz, and Kreuzer, *Stretching Response of Discrete Semiflexible Polymers*, *Macromolecules*, 2003 with QM corrections for backbone stretching given in (Hugel et al., *Highly Stretched Single Polymers: Atomic-Force-Microscope Experiments Versus Ab-Initio Theory*, PRL 2005) which become relevant at these extremely high forces and would give them quantitative contour and persistence length values for their force-extension curves.

We agree with the reviewer that our interpretation of the molecular spring constant was not appropriate. We apologize for this and we have removed all the data and discussion concerning k_m , which were not needed for the message of the manuscript. Moreover, we have re-analyzed all our data using an extensible WLC model, which includes the effect of directly stretching the chain backbone (*Angew. Chem. Int. Ed.* 39, 3212–3237 (2000)) and thus fits better our data even at forces higher than 1 nN, as indicated by the reviewer. We also provide now contour length data rather than rupture length, as it is a more rigorous parameter to describe our interaction as the reviewer pointed out. Though this extensible WLC model has been questioned in Hugel *et al.* and Livaduru *et al.*, it seems to fit correctly our data even at high forces. We were not able to implement such corrected models but as WLC+ model fitted our data we feel that this level of analysis is fine, particularly given the objective and message of our paper (catch bond).

In Fig. 3a it is unclear which retraction velocities gave which data point cluster, these should be color coded to show which retraction velocity was used. The large spread of high force rupture forces if not acquired from a single cantilever may be cantilever calibration artifacts and should be discussed.

We feel such color version of the DFS plot, as well as the Bell Evans fit, would make the main figure more difficult to read. But, we now provide a color-coded version of a DFS plot (3 cells merged) in figure 2 below.

Figure 2. Dynamic force spectrum on three merged SpsD cells and color-coded depending on the retraction velocities.

In addition, we doubt the tiny differences observed in between cells (see new figures 1, 2 and supplementary figures 1, 2 and 4) arise from calibration issues, though always possible. We feel this could be rather related to cell variability and AFM tip chemistry, as now discussed in lines 99-101.

Lastly, the rupture forces around 60 pN with low force loading rates are most likely non-specific interactions. Given that a live bacterium's surface is not blank, a discussion of how specific from non-specific events were discerned should be provided. Especially as the target molecule fibrinogen is rapidly coating surfaces and very sticky. Low (~ 100 pN) force unbinding events appear in the negative control in Fig 1. Yet, they are discussed as specific "weak adhesion force" and evidence for force activation in the dynamic force spectra. Likely, these low-force events are unspecific attachment of bacterium to surface, or fibrinogen. Please resolve this unclarity with reference to the control experiments.

Despite the force-distance curves profiles show that the nature of this small peak is clearly specific. We agree with the reviewer that it might be an « artefact » derived from the lack of orientation of Fg molecule on the AFM tip or other kind of interactions between bacterial membrane and Fg, as now mentioned in lines 108-113. As we can also observed in our control experiments weak forces in the range, it seems reasonable that this interaction could arise from some kind of uncontrolled interaction with the bacterial membrane so that we have decided to not consider them.

Secondly the attachment strategy used to anchor the Fg target by NHS/EDC chemistries is not site-specific meaning multiple pulling geometries of Fg are possible (e.g. from N- or C-terminus, or even both termini). Previous work cited has shown a decisive influence of pulling geometry in rupture force for SpsD's homolog SdrG (60 pN non-natively, over 2000 natively) The possibility of such non-native low force geometries occurring here should be discussed, these could also be the cause of low force interactions.

We totally agree with the reviewer concerning the influence of the pulling geometry as elegantly demonstrated by Milles *et al.* (Science 359, 1527–1533 (2018)). See above.

The statement that Bell-Evans or Friddle models do not apply here is not substantiated by

the data shown. Indeed, the authors do not even try to fit these models here. A Bell-Evans fit to the rupture force histogram at high forces e.g. in Fig. 3b or Supplementray Figures 1 and 2, should be performed, judging by the shape of the distribution it should converge and yield a zero force off-rate and a distance to the transition state that could also be used to compare to such parameters acquired for six other DLL adhesins referenced in previous work.

We have now revised our vision of the DFS plot: our data in the high force regime ~ 1.5 nN are well fitted by the Bell Evans model, as shown in new figure 3a.

Have the authors considered fitting explicit catch bond models directly to their data, such the sliding-rebinding model as in Rakshit et al. PNAS 2012 for cadherins?

We have attempted to fit our data with previous theoretical formulations on catch bonds, notable the sliding-rebinding model, the deformation model and the one-bound state, two unbinding pathways model, but without any success. It is actually not surprising that our catch bond cannot match anticipated models of the literature as it occurs at forces orders of magnitude higher (nN range) than those reported for previously investigated purified systems (pN range). Consequently, we hypothesize that the SpsD catch bond may involve an unconventional model where propagation and direction of the applied force play central roles, as already proposed by Gaub team (see e.g. Nano Lett. 15, 7370–7376 (2015), J. Am. Chem. Soc. 141, 14752–14763 (2019)) and discussed in lines 185-212.

The discussion part proposes that the DLL mechanism only completes, or rather deforms to form a catch bond, upon force application. However, previous crystallographic and thermodynamic studies for these adhesins have shown that the bound state of a target inside the binding cleft between N2 and N3 domain, and a closed “latch” is achieved upon, and even required for, target binding, most notably: Bowden et al., Evidence for the "dock, lock, and latch" ligand binding mechanism of the staphylococcal microbial surface component recognizing adhesive matrix molecules (MSCRAMM) SdrG, JBC 2008 Thus, it seems unlikely, all the while possible, that mechanical forces enhance this binding process, definitive evidence for this hypothesis would require to show that force application enhances the complexes' kinetic on-rate.

We apologize if our discussion was misleading. We hypothesized that upon DLL interaction is formed, under tensile force the molecular complex is stretched, inducing structural changes that can lead to different lifetimes of the bond.

The aspect of force causing major conformational changes to create a catch-bond is worth of debate, indeed the work referenced for the formation of backbone hydrogens bonds showed no discernible or large conformational changes in the adhesin:ligand complex in MD simulations for SdrG – which clashes with the proposed deformation model. However, these results only apply to SdrG, SpsD may be a homologous, albeit mechanistically very different adhesin.

We have now reformulated our statements in lines 185-212 to clarify that we do not claim that force-induced major conformational changes lead to a catch bond behavior. However, once the DLL interaction between SpsD and Fg is achieved upon binding, the extent of stretching force applied before clamping the bond might lead to some structural modifications of the complex, and causing the catch-slip bond transition we observe.

The methods are incomplete. Constructs and strains used should be given in exact sequence, Vectors (if used) and strains used should be specified exactly with databank accession numbers, protein sequences, plasmids etc. used should be given in great detail to enable reproduction of this work. Currently, it is unclear which sequences were used here.

We have now indicated this information in the Methods section.

Unfolding peaks preceding receptor ligand unbinding in curves are never discussed, the sequence of the gene would not suggest that these be present. To these arise out of the stretched Fg, or additional interactions? The peaks preceding complex rupture in curves as seen in curves in Fig 1b should be analyzed for repeated contour length increments that would indicate a specific domain unfolding and discussed. The reviewer is missing that we only observed this small peak preceding the strong interaction in single-cell experiments, where the whole bacteria is brought into contact with a Fg coated surface. Thus, the contact is bigger and consequently can arise more unwanted interactions. However, we barely observed the presence of this small peak in single-molecule. As discussed above this small peak could correspond to unspecific interactions between bacterial membrane and Fg.

Further comments in order of appearance in the manuscript:

Abstract:

The abstract claims that the only microbial catch bond mechanism discussed in molecular detail is FimH. Nord et al. PNAS 2017 have dissected catch bonding in rotor/stator interactions of the bacterial flagellar motor.

Indeed, Nord *et al.* have reported a mechanosensitive complex in the flagellar motor and suggest it behaves as a catch bond. The work doesn't perform force clamp experiments, which we feel is the most unambiguous way to demonstrate a catch bond, but they suggest a catch bond behavior by modifying viscous load and by extraction of kinetic constants. In addition, the flagellar motor is involved in bacterial motility rather than bacterial adhesion. In this work we are focusing on cell adhesion processes, so that we refer to demonstrated catch bonds only in the context of cell-surface interactions.

The wording of "holy grail", both stylistically and factually seems out of place here. Especially, since catch-bonding mechanisms have already been resolved in pathogens such as in the case of FimH as cited by the authors.

Please specify how catch bonding mechanisms will lead to antibacterial strategies. We have tempered our claims.

Line 46 ff.

Adhesion ligand systems such as Titin:Telethonin, Streptavidin:Biotin, or the Cohesin Dockerin Type III system and its variants with similar functions are far above these values, 400 to over 1000 pN. (Bertz et al., PNAS 2009; Sedlak et al., Sci. Adv. 2020; Bernardi et al. JACS 2019). 250 pN seems a low value for typical interactions especially considering more recent work.

We have reformulated our thoughts in lines 40-43 and have included those references pointing to recent complexes involving much higher forces than previously reported.

Line 55:

Please expand and reference the occurrence of these “extreme mechanical forces”

We have now provided a reference for this statement (Persat *et al.* “The mechanical world of bacteria”, *Cell* **161**, 988–997 (2015)).

Results:

Line 84 ff.

“that its folded length is ~20 nm”

How do the authors arrive at that distance, please provide a reference or a detailed explanation? How would the 1031 amino acids of the sequence result in just 20 nm of effective folded length? The value of 1031 amino acids used to calculate the contour length is incorrect. Firstly, the N1 domain is not involved in ligand binding so it’s around 130 amino acids do not count into the contour length. Secondly, raw sequence of the SpsD gene shown in Fig. 1a is also processed during export and sortase-mediated attachment, which should be discussed as it will change the contour length of the protein - it will likely be shorter in its mature form. Please recalculate these values.

We thank the reviewer for the correction; we have now recalculated the expected length of SpsD, which can be stretched, thus excluding the SS region and the N1 domain (see lines 85-91).

The authors mention the length of the Fg used, but would also need to justify where it was attached and how that contributes to its expected contour length.

Fg was grafted to the AFM tip through amine groups using NHS/EDC chemistry, as mentioned in the Methods section. Since this chemistry does not provide control of the orientation, we cannot predict to which extent Fg is contributing to the contour length. However, as stated in lines 89-91 and because our data do not allow to discriminate between both adhesion and ligand contributions, we consider that both Fg and SpsD could be stretched and contribute to the contour length.

Please provide a reference for the contour length per amino acid.

We have now provided references for the contour length per amino acid (Park et Muller, *Methods Mol. Biol.* Clifton NJ 1271, 173–185 (2015)).

In Figure 1 the rupture distance but not the contour length is given, please fit appropriate models to the force extension curves to arrive at a contour length, or at least provide a rationalization or estimate of the contour length of the stretched complex.

We have reanalyzed all our data using this time the extensible WLC model and substitute the rupture length by contour length histograms. See above for more details.

Fig.1 what retraction velocity was used to acquire the data shown?

The data shown are obtained under our standard conditions, *i.e.* a retraction velocity of 1 $\mu\text{m/s}$ as now mentioned in the caption.

Line 95: Please provide references and details for how a DLL mechanism was shown for

this adhesin. High forces alone are not an argument for the DLL mechanism, it is conceivable – yet unlikely given SpsD’s high homology to established adhesins like SdrG - that this is a different or at least varied binding mechanism.

Pietrocola *et al.* (PLoS ONE 8, e66901 (2013)) constructed Δ LL and trench mutants of the N2N3 subdomains of the A domain of SpsD and showed that binding to the γ -chain of Fg was completely abrogated in these cases, demonstrating the interaction between SpsD and Fg occurs by DLL. This can also be highlighted by the close sequence identity of SpsD with FnBPs for which the binding to Fg is known to be mediated by the DLL (Kean, F.M. *et al.*, Molecular Microbiology (2007) 63(3), 711–723), as it has also been demonstrated for the closely related ClfA and ClfB by X-ray crystal structure (Ganesh, V.K. *et al.*, J Biol Chem. (2011);286(29):25963-72). The high forces and the dynamical behavior reported in our manuscript are obviously not a direct proof of a DLL mechanism but reinforce / confirm such strong hypothesis, especially because they match perfectly those previously reported for FnBPs and Clfs, staphylococcal adhesins known to involve DLL mechanism while binding to Fg (see cited references in the manuscript).

Please provide an explanation for the unbinding force differences of around 400 pN between Fig. 1 and Fig. 2 for essentially the same system unbinding. Could cantilever calibration errors play a role here?

We believe differences in adhesion forces from single-cell and single-molecule experiments could be due to variability in bacterial cells and chemistry used for tip and surface functionalization (see lines 99-101), but could also be somehow interfered by differences in cantilever spring constant, though less probable. Indeed, below we have added 2 tables with all the probed cells in each set of experiments and the corresponding cantilever spring constant used, showing that there is no direct correlation between this parameter and the mean adhesion strength we observed on cells.

Table 2. Mean adhesion force value and calibration value for SC experiments for each cell.

Cell	Mean adhesion force (pN)	Cantilever spring constant (N/m)	
1	1900	0.12	Cantilever1
2	1899	0.12	
3	2012	0.12	
4	1954	0.12	
5	1908	0.12	
6	1727	0.095	Cantilever2
7	1686	0.095	
8	1624	0.095	
9	1747	0.095	Cantilever3
10	1863	0.095	
11	1836	0.099	Cantilever4
12	1862	0.099	
13	1717	0.099	
14	1719	0.099	
15	1725	0.099	

Table 3. Mean adhesion force value and calibration value for SM experiments for each cell.

Cell	Mean adhesion force (pN)	Cantilever spring constant (N/m)	
1	1537	0.022	Cantilever1
2	1458	0.022	
3	1484	0.022	
4	1391	0.022	Cantilever2
5	1604	0.022	
6	1589	0.022	
7	1604	0.032	Cantilever3
8	1704	0.017	Cantilever4
9	1551	0.051	Cantilever5
10	1542	0.051	
11	1653	0.051	

The authors argue that clustering of adhesins on the bacterial surface show that single complexes unbind in their data. This argument should be refined as receptor clusters make binding events of multiple adhesins leading to higher forces more likely. The rupture force populations at around 3000 pN could be the source of this cluster binding, please discuss. We agree with the reviewer that presence of adhesin cluster could induce multiple binding events. However, as long as the curve shows a single and well defined specific peak we don't have any scientific/technical reason that allow us to distinguish if this big forces arise from multiple bindings or single interactions.

Line 148:

With several catch bonds reported (even directionally asymmetric ones such as by Huang et al. Science 2017 for vinculin/actin) or Notch-Jagged (Luca et al. Science 2016) I would hardly characterize them as a "controversial" phenomenon.

We agree with the reviewer and have removed the term from the discussion.

Please provide statistics for the experiments, curve attempts, successful single interactions, measurement time, and show more raw force extension curves and force clamp curves in the supplement.

We have included more raw curves in main figures and supporting information for all the experiments and added more detailed information in materials and methods and captions.

Cantilever spring constants may influence the system under investigation. Please provide the exact spring constants measured from cantilevers associated with each measurement in the supplement. Please provide a detailed calibration method including InvOLS determination.

Please see above.

Reviewer #3 (Remarks to the Author):

Referee Report: “Force-clamp spectroscopy identifies a catch-bond mechanism in a Gram-positive pathogen,” by Mathelié-Guinlet et al. In their paper “Force-clamp spectroscopy identifies a catch-bond mechanism in a Gram-positive pathogen,” Mathelié-Guinlet et al. investigate the dock, lock, and latch (DLL) interaction between staphylococcal surface protein SpsD and fibrinogen using atomic force microscopy in the dynamic force spectroscopy mode and in force-clamp spectroscopy mode. In dynamic force spectroscopy, authors find unusually high bond rupture forces (1 – 2 nN) above force loading rates of 1,000 pN/s and much lower rupture forces (approx. 60 pN). Using force clamp spectroscopy, the authors show, that forces up to 1,100 pN enhance the bond strength and increase the mean bond lifetime (up to approx. 6 s), while at forces above 1,100 pN the bond weakens and the bond lifetime decreases again. Owing in large extent to the use of the force clamp method (Figure 4), this paper represents indeed the first study which unambiguously and convincingly demonstrates catch-bond behavior for adhesion molecules of Gram-positive bacteria. The study is well done and the paper well written. The study represents a major breakthrough in mechanobiology and provides a better understanding of infection mechanisms of Gram-positive bacteria. The paper should therefore clearly be published by Nature Communications, after the authors, have addressed the following (minor) shortcomings of their manuscript:

We thank the reviewer for his / her very positive feedback. Below, we have answered all the concerns, which helped us to improve our manuscript.

1. As mentioned above, owing to the force clamp method used, this is the first paper clearly and unambiguously demonstrating catch-bond behavior in adhesion molecules from Gram-positive bacteria. Nevertheless, it is not the first AFM study pointing to catch bond behavior in such adhesins. The authors should make this clear in the introduction and discussion and reference previous studies which observed catch-bond like characteristics in other receptor ligand systems of Gram-positive bacteria. An overview can be found in reference 3 of this manuscript as well as in ACS Nano, 13, 7155-7165 (DOI:10.1021/acsnano.9b02587)

The reviewer is missing that we claim the first direct demonstration of catch bond on living bacteria, while still citing previous studies that pointed to such behaviors. In the suggested reference, there is indeed no direct evidence of a catch bond. All the work is based on dynamic force spectroscopy experiments, which only show force enhancement with mechanical stress and thus suggests a potential catch bond behavior. Again, in those works, there is no direct or indirect measurement of the lifetime of the bond, consequently there is no direct proof of a catch bond.

2. Fig. 4d and line 130: “The bond survival probabilities at almost each force clamp were described by a single exponential decay.” The curves at 1000 pN (blue) and 1100 pN (green) clamp force in Fig. 4d clearly show multi-exponential decay. The other curves also show deviations from a single exponential behavior. The authors should include the fits of $N(t)$ which were used to determine τ in Fig. 4d. This would help the reader to judge whether the decay is single or rather multi-exponential. As mentioned, the curves at 1000 pN (blue) and 1100 pN (green) clamp force show multi-exponential decay. A bi-exponential

kinetic model might be more suitable here. Examples of such models can be found e.g. in *ACS Nano*, 6, 1314-1321, (DOI: 10.1021/nn204111w) and in *Angewandte Chemie Int. Ed.*, 58 (29), 9787-9790 (DOI: 10.1002/anie.201902752). Please show fits, and consider a multi-exponential fit at 1000 pN and 1100 pN. The authors should also change Fig. 4e accordingly, by giving both lifetimes, when a bi-exponential kinetic model fits the data better than a single exponential model, and discuss possible reasons for the bi-exponential kinetics in the paper.

To confirm more explicitly our catch bond behavior, we have extended the statistics obtained for all forces, especially those at lower forces (800 and 900 pN). The revised manuscript now includes $n = 108, 109, 116, 124, 135$ and 140 events for $F = 800, 900, 1,000, 1,100, 1,200, 1,300$ pN respectively, recorded on 11 independent cells from 6 independent cultures (see new figure 4). After extending these statistics, we still observe a clear catch – slip transition with maximum lifetime at $F = 1000-1100$ pN. A non-parametric Kruskal Wallis test has confirmed the significant differences between lifetimes obtained at 800, 900, 1200 and 1300 pN with those obtained at 1000 and 1100 pN with $p < 0.0001$, demonstrating the catch-slip transition at those later forces for which no significant differences were reported.

As the fits are concerned, for some of the clamping forces, we attempt to perform a double exponential fit, as depending on the time regime the decay was apparently not unique (see especially 900 pN). However, for most of our data, a single exponential decay was well describing the survival plots and is better consistent with the DLL mechanism involved in the SpsD-Fg interaction. Indeed, a double exponential decay would involve a second bound state that it is unlikely to occur in DLL. Consequently, we have extracted lifetime values only from single decay fits as presented in new figure 4.

3. Lines 114 – 120 and Fig. 3C. The author report differences in the molecular spring constant (k_m) in the different force regimes and conclude that “This strongly supports a deformation model in which tensile force induces a switch from a soft, weak binding conformation, to a stiff, strong-binding conformation, resulting in a mechanically stabilized adhesin-ligand complex.” The molecular spring constant proteins and polypeptides is highly non-linear and increase in molecular spring constant with force is a normal behavior of these molecules. This holds true for any protein. Thus, this observation does not explain or support catch-bond behavior. The relevant parameter here is not the molecular spring constant, but the stretch modulus (S) which can be obtained by fitting an extensible worm like chain model to the force extension curves, as it is commonly done in the field. The authors should show S rather than k_m , or remove lines 114-120 and Fig. 3C from the manuscript.

We agree with the reviewer that our interpretation of the molecular spring constant was not correct, in light of those explanations. Indeed, at high forces $\sim 1,500$ pN, interaction is likely to be dominated by the energetic regime rather than the entropic one due to high stretching of the chain backbone. Consequently, our definition of k_m and its subsequent interpretation were definitely wrong. We have re-analyzed all our data using an extensible WLC model (*Angew. Chem. Int. Ed.* 39, 3212–3237 (2000)), which includes the effect of directly stretching the chain backbone and thus fits better our data. Nonetheless the stretch modulus extracted from this model do not actually provide any additional information to support the main message of the paper. We have thus decided to remove all the data and discussion related to the molecular spring constant / stretch modulus of the complex.

4. Fig. 2 and Fig. 3: The authors show data from only 3 cells in these figures and move the data from three more cells to the Extended data. I would recommend to merge data from Fig. 2 and 3 and Extended Data and show data from all six cells in the main figures (2 and 3).

We consider that showing 3 cells in main figures and 3 more in supplementary is the best way to show the data. By merging 6 cells in a single histogram the reader would have difficulties in getting clear information from the graphs.

5. Lines 85/86: „Assuming that (i) the processed mature SpsD adhesin comprises 1,031 residues, (ii) that its folded length is ~20 nm“

We thank the reviewer for the correction; we have now recalculated the expected length of SpsD, which can be stretched, thus excluding the SS region and the N1 domain (see lines 85-91).

6. Line 88: “This suggests that 87 SpsD and, to some extent Fg, are being stretched upon pulling the cells away from the Fg-surfaces.” If ruptures occur at 351 ± 62 nm, why is Fn also stretched? Do you assume that SpsD remains partially folded at these forces?

The interaction between SpsD and Fg occurs through the N2 and N3 domains, meaning that N1 wouldn't be directly involved in the binding and subsequently might not be stretched. Since our contour length data do not allow us to discriminate which of the two proteins are actually stretched and to which extent, we reason that both molecules could contribute to the rupture values.

7. Line 155: Typo: my=> by

The typo has been corrected.

8. Fig. 4a/line 353: the black and blue curve are difficult to discriminate in the printed manuscript. Consider using another color.

We have changed the blue color.

9. Line 355: Typo: ... curves obtained at different loads, and...

The typo has been corrected.

10. Line 414: How were the bacteria immobilized on PS?

A drop of 50 μ L of bacteria was directly deposited on the polystyrene dish, let to stand for bacteria to adhere and then rinsed with PBS to remove unattached bacteria.

REVIEWER COMMENTS

Reviewer #2 (Remarks to the Author):

The revised version of the manuscript by M Mathelié-Guinlet, F Viela et al. has much improved and addresses the main concerns raised in my initial review. Congratulations to the authors in establishing the catch bond behavior in this pathogen adhesion system.

The manuscript is ready for publication now, given that the minor corrections given below are addressed, which I am confident the authors can do quickly. Thus, I do not see a need to review the manuscript again.

Please add all the additional plots used in the rebuttal to the supplement. Especially the table of spring constants determined and fit parameters. This is crucial information and should not be withheld from the manuscript.

Regarding analysis of force loading rate:

using the final part of the force extension curve directly before complex rupture is a valid method to extract this value. However, the statement in the rebuttal that "the loading rate is not constant anymore due to elastic stretching of the complex, which, in turn, modify the energetic properties of the bound."

Should reconsider that the entropic elasticity in these curves comes from the polymers (like flexible protein linkers) it seems debatable that the protein-protein complex itself is "elastic", also given its size in comparison the rest of the protein.

Regarding cell/force variability:

Please explain explicitly how surface chemistry will modify the behavior of the system investigated. Other points to consider would be the variability/uncertainty of AFM cantilever calibration can easily be in the range of 10%, as per: (Brand et al., Comparing AFM cantilever stiffness measured using the thermal vibration and the improved thermal vibration methods with that of an SI traceable method based on MEMS, Measurement Science and Technology 2017). This effect should always be kept in mind when comparing AFM datasets from different cantilevers, and especially here were no InvOLS was determined? If the InvOLS was determined please give this in the methods.

Likely it is the origin of the variability observed here. Secondly, I do agree that the stiffness of different bacteria will have an effect on the effective cantilever spring constant which may influence complex rupture behavior, which may be worthy of discussion here. Please revise in line 104.

Regarding methods:

Complete accession numbers for strains and the exact sequence of the protein studied here with links into standard databases (genbank, uniprot etc.) would greatly help if this work is to be reproduced at a later time.

Please discuss the values extracted from the Bell Evans model fit, do they correspond to previous work, what are their uncertainties? Please specify these values in the main text and not just in the figure caption. What are the errorbars in the Bell-Evans fit.

Please specify exactly how the e-WLC (please explicit state the equation, to avoid confusion as to what the e-WLC is) model was fit, which values were obtained or used to achieve the fit, please show a fit on a force-extension curve to demonstrate that it converges.

minor comments:

line 92: why is the S-region not exposed? Is this in reference to the SpsD gene?

Line 185: might want to add non-covalent, biomolecular covalent interactions are stronger.

Figure 3 title: please revise "is activated" to "is enhanced" as in the main text.

Figure 3: (b) "stress" is ambiguous, please change to force loading rate which is what is shown here.

Reviewer #3 (Remarks to the Author):

In the revised manuscript, the authors have addressed most of the concerns of the reviewers. This has significantly improved the quality of the manuscript. However, one major concern regarding the analysis and interpretation of the data has not been adequately addressed by the authors: None of the survival probability vs. time curves shown in Figure 4C, except for the 800 pN curve show the single exponential behavior claimed by the authors, which would correspond to a straight line in the semi-logarithmic representation in Fig 4C. Instead, all datasets, except the 800 pN one, deviate markedly from such a behavior at long survival times. The only data set which follows single exponential decay, the 800 pN one is also the one which is furthest away from the transition force from slip bond to catch bond (1,100 pN).

In their rebuttal, the authors argue that "a double exponential decay would involve a second bound state that it is unlikely to occur in DLL". However, just assuming that something is "unlikely to occur" does not justify ignoring obvious features of the experimental data. Although all experimental data sets except the 800 pN one clearly show signs of multi exponential decay, especially the 1,000 pN and the 1,100 pN data sets, in their rebuttal the authors state, that a single exponential model fits the data better than multi exponential models. However, for all fits shown in Fig 4C, the fit-range ends and the fits are truncated before the data sets become multi exponential. Therefore, the fits do not capture the multi exponential features of the data, which are however clearly visible even by eye.

In the present form, I cannot recommend publication of the manuscript. The authors have to use fit ranges which cover the entire data sets and which are not truncated at long survival times. They should give chi-square values for their fits, so that the reader can quantitatively judge and compare the quality of the fits (not just by eye). Finally, the authors clearly have to consider a bi-exponential kinetic model, even if they consider bi exponential decay unlikely based on their assumptions and hypothesis on the DLL bond rupture mechanism. The data presented clearly contradicts single exponential kinetics.

Reviewer #2 (Remarks to the Author):

The revised version of the manuscript by M Mathelié-Guinlet, F Viela et al. has much improved and addresses the main concerns raised in my initial review. Congratulations to the authors in establishing the catch bond behavior in this pathogen adhesion system. The manuscript is ready for publication now, given that the minor corrections given below are addressed, which I am confident the authors can do quickly. Thus, I do not see a need to review the manuscript again.

We thank the reviewer for his very positive feedback. We have answered point-by-point all his concerns below and, accordingly, in the revised manuscript.

Please add all the additional plots used in the rebuttal to the supplement. Especially the table of spring constants determined and fit parameters. This is crucial information and should not be withheld from the manuscript.

We have now included in the Supplementary Information (i) the table of the fitting parameters for the catch bond behavior, (ii) the tables of the cantilevers' spring constants used in this work and (iii) the color-coded DFS plot showing data obtained with different velocities on three representative cells.

Regarding analysis of force loading rate:

using the final part of the force extension curve directly before complex rupture is a valid method to extract this value. However, the statement in the rebuttal that "the loading rate is not constant anymore due to elastic stretching of the complex, which, in turn, modify the energetic properties of the bound." Should reconsider that the entropic elasticity in these curves comes from the polymers (like flexible protein linkers) it seems debatable that the protein-protein complex itself is "elastic", also given its size in comparison the rest of the protein.

We agree with the reviewer comment.

Regarding cell/force variability:

Please explain explicitly how surface chemistry will modify the behavior of the system investigated. The surface chemistry used should not change the behavior of the system investigated. But, due to EDC/NHS chemistry, it might be that the ligand interacting with the adhesin is not located at the extreme end of the tip, thus leading to pulling geometry effects for instance.

Other points to consider would be the variability/uncertainty of AFM cantilever calibration can easily be in the range of 10%, as per: (Brand et al., Comparing AFM cantilever stiffness measured using the thermal vibration and the improved thermal vibration methods with that of an SI traceable method based on MEMS, Measurement Science and Technology 2017). This effect should always be kept in mind when comparing AFM datasets from different cantilevers, and especially here where no InvOLS was determined? If the InvOLS was determined please give this in the methods. Likely it is the origin of the variability observed here. Secondly, I do agree that the stiffness of different bacteria will have an effect on the effective cantilever spring constant which may influence complex rupture behavior, which may be worthy of discussion here. Please revise in line 104.

We agree with the reviewer that (i) the uncertainty on the calibration of the cantilever spring constant, and (ii) AFM datasets obtained with different cantilevers (thus of different spring constants) could

explain the variability of forces observed in this work, along with our privileged hypothesis concerning cell variability. This has been mentioned in the revised manuscript, lines 102-105.

Regarding methods:

Complete accession numbers for strains and the exact sequence of the protein studied here with links into standard databases (genbank, unipot etc.) would greatly help if this work is to be reproduced at a later time.

The details have been provided in the revised manuscript, in the Methods section, lines 241-247.

Please discuss the values extracted from the Bell Evans model fit, do they correspond to previous work, what are their uncertainties? Please specify these values in the main text and not just in the figure caption. What are the errorbars in the Bell-Evans fit.

We have now included a comment in the revised manuscript concerning the parameters obtained by the Bell Evans model, on lines 131-135. We obtained $x_{\beta} = 0.09 \pm 0.02$ nm and $k_{\text{off}} = 5 \times 10^{-13} \pm 4 \times 10^{-12} \text{ s}^{-1}$. While x_{β} is comparable to the one obtained by Milles *et al.* for SdrG binding to Fg through the DLL ($x_{\beta} = 0.051$ nm), our k_{off} is two orders of magnitude lower ($k_{\text{off}} = 9.2 \times 10^{-11} \text{ s}^{-1}$), though remaining very low as for SdrG.

Please specify exactly how the e-WLC (please explicit state the equation, to avoid confusion as to what the e-WLC is) model was fit, which values were obtained or used to achieve the fit, please show a fit on a force-extension curve to demonstrate that it converges.

This information has been added to the Methods section, on lines 289-292.

minor comments:

line 92: why is the S-region not exposed? Is this in reference to the SpsD gene?

We apologize for this error but we actually meant the sorting signal, SS, that is buried should not be exposed. This has been fixed in the revised manuscript.

Line 185: might want to add non-covalent, biomolecular covalent interactions are stronger.

Figure 3 title: please revise "is activated" to "is enhanced" as in the main text.

Figure 3: (b) "stress" is ambiguous, please change to force loading rate which is what is shown here.

These three comments have been taken into account in the revised manuscript, as required.

Reviewer #3 (Remarks to the Author):

In the revised manuscript, the authors have addressed most of the concerns of the reviewers. This has significantly improved the quality of the manuscript. However, one major concern regarding the analysis and interpretation of the data has not been adequately addressed by the authors: None of the survival probability vs. time curves shown in Figure 4C, except for the 800 pN curve show the single exponential behavior claimed by the authors, which would correspond to a straight line in the semi-logarithmic representation in Fig 4C. Instead, all datasets, except the 800 pN one, deviate markedly from such a behavior at long survival times. The only data set which follows single exponential decay, the 800 pN one is also the one which is furthest away from the transition force from slip bond to catch bond (1,100 pn).

In their rebuttal, the authors argue that “a double exponential decay would involve a second bound state that it is unlikely to occur in DLL”. However, just assuming that something is “unlikely to occur” does not justify ignoring obvious features of the experimental data. Although all experimental data sets except the 800 pN one clearly show signs of multi exponential decay, especially the 1,000 pN and the 1,100 pN data sets, in their rebuttal the authors state, that a single exponential model fits the data better than multi exponential models. However, for all fits shown in Fig 4C, the fit-range ends and the fits a truncated before the data sets become multi exponential. Therefore, the fits do not capture the multi exponential features of the data, which are however clearly visible even by eye.

In the present form, I cannot recommend publication of the manuscript. The authors have to use fit ranges which cover the entire data sets and which are not truncated at long survival times. They should give chi-square values for their fits, so that the reader can quantitatively judge and compare the quality of the fits (not just by eye). Finally, the authors clearly have to consider a bi-exponential kinetic model, even if they consider bi exponential decay unlikely based on their assumptions and hypothesis on the DLL bond rupture mechanism. The data presented clearly contradicts single exponential kinetics.

We apologize if the reviewer feels that we did not properly address this issue, this is likely due to lack of details on how we chose to fit our data. We have included a more detailed explanation in the text clarifying our criteria to fit our data, in lines 155-157 and 163-167, as well as a table summarizing the fitting parameters (notably R^2 and χ^2) in the supplementary information.

We are confident that our single exponential decay fits reflect and adjust to the real mechanism involved in the SpsD-Fg interaction because of the following reasons. (1) There is no experimental evidence on the DLL mechanism reporting, not even suggesting, a two bound states in previous literature. We thus believe that the mathematical model reflects the real DLL interaction. (2) We chose to fit the survival probabilities on a time range that encompasses most of the raw lifetime data, depending on the clamping force. (3) Bi-exponential decays are not fitting (they do not even converge) our data in most of the cases.

In figure R1 below, we have plotted the Kernel density plots to show the distribution of our raw lifetime data, depending on the clamping force. We observe that at $F = 900, 1200$ and 1300 pN most of the data are in a range between 0 and 3 s, while at $F = 1000-1100$ pN the range extends up to 4 s. Moreover whatever the clamping force is, the highest density of data is observed between 0 and 2 s and drastically drops after. Therefore, we believe it is fair not to force a fitting to *e.g.* a bi-exponential decay on the full time range, only for the fact that plots have an apparent transition at higher times that might be due to some residual experimental data not reflecting the real mechanism.

Figure R1. Kernel density plots of the lifetimes of the SpsD-Fg bonds depending on the force clamp.

Yet considering the suggestion of the reviewer we have done many attempts and efforts to fit our data to the bi-exponential decay model. In 3 out of 5 clamp forces, the fits do not converge after trying several iterations (from N=400-1000) (Fig. R2 and Table 1). At $F = 1300$ pN, though the fit seems to converge the standard deviation of the time is several orders of magnitude higher than the centered value, indicating that the bi-exponential decay do not properly fit our experimental data.

In summary, we are not only convinced that our system involved in a DLL mechanism should not theoretically involve two binding states, *i.e.* the bound survival probability should not be fitted by a two exponential decay, but also we chose to fit our data on a time range that reflects most of the data observed, showing on the survival plots the full time range out of honesty.

Table 1. Fitting parameters obtained by fitting the survival plots at different clamping force by a bi-exponential decay.

	Max. number of iterations	Convergence	t_1 (s)	t_2 (s)	χ^2	R^2
900 pN	400	Yes	0.157 ± 0.013	1.259 ± 0.023	$3,5 \cdot 10^{-4}$	0.991
1000 pN	400 to 1000	No - overparametrization				
1100 pN	400 to 1000	No - overparametrization				
1200 pN	400 to 1000	No - overparametrization				
1300 pN	400	Yes	0.777 ± 211	0.777 ± 194	$3,8 \cdot 10^{-4}$	0.991

Figure R2. Bond survival probabilities for the SpsD-Fg interactions, measured at 5 different clamping forces. Bi-exponential fits are presented as dashed lines with corresponding color. The number of iterations used and the fit convergence is also mentioned.

REVIEWER COMMENTS

Reviewer #2 (Remarks to the Author):

The revisions given by the authors leave no major concerns from my side. The manuscript is ready for publication if the very minor comments below are addressed. Again, congratulations to the authors. An assessment of R3 comments can also be found below.

Minor comments:

Line 135-136:

Is the affinity of SpsD known? The affinity of DLL adhesins is in the micromolar range, so not exceptionally high. The koff rates here do not relate to the bulk affinity of the system as they are measured under force. In general, single-molecule mechanics and bulk binding kinetics are not correlated (e.g. systems of much higher affinity have lower mechanical stabilities and a much faster koff0, notably Streptavidin:Biotin). Please rephrase.

Figure S1 should be more explicitly labeled (exact surface/cantilever bound molecules/exact cell conditions)

Figure S6, please provide a zoom in on the force traces, the constant force plateau is barely visible to judge the stability of the force clamp (i.e. please explicitly show plots of the constant force region in the range of the respective clamping force). Currently the features in the trace are not recognizable. Please also show the extension vs. time traces to highlight the speed of the feedback on the cantilever position, as this was a major point of discussion in the initial review.

The explicitly shown fit of the e-WLC model to force-extension data with explicit parameters fitted is not included yet, it must be added in the final version.

It would be advisable to add all data shown in response to R3 to the supplement. What is the x-axis in and the unit of the heatmap shown in Fig. R1? These plots are somewhat difficult to decipher. The raw data of the force clamps should be published alongside the manuscript so that in the future the discussion with R3 might be resolved through different model fits.

In regard to the major concerns raised by R3:

R3 raises a valid point, I think all the mathematical requests given can be easily addressed (chi-squared values for the fits, discussion of truncations in the data in Fig. 4C). The revision if it addresses R3's requests and all the minor corrections and technical as well as obvious requests (strain accession numbers are still missing for example) will cover the minimum required to publish the work.

The question of the exponential decay remains. The non-single-exponential decay in Figure 4C raises concerns as to what is measured here, and is highly unexpected for a single interaction dissociating under force. It is also concerning that long interaction lifetimes (which are among the key characteristic of a catch bond) show the strongest deviation. The explanation given for this deviation by the authors is not satisfying, R3 is right. To fulfill R3's requests the authors should show many specific force-time curves for all lifetime, as they have done now. It would be advisable to show even more, with explicit zoom ins of the the constant force regime when clamping, and grouped by lifetime regimes regimes (e.g. 1s , 2s, 3s, 4s etc., independent of clamping force) which could already help to discern qualitative differences in these curves.

I see three ways to satisfy R3's requests and resolve this problem:

- careful re-examination of the data, many traces of force clamping should be shown. Fitting of a simplified bi-exponential model as two individually fitted exponentials (the regions could be initially chosen by eye) for Fig. 4C as requested by R3 and a discussion why the decay is not single-exponential (which could be caused by non-specific tethering geometries, or multiple SpsD/fibrinogen interactions, as raised as a concern in my initial review). This is probably the easiest option, albeit

maybe not the most satisfactory for R3.

- the authors cede the point that uniquely single receptor-ligand interaction is the basis for the behavior (which would make the bi-exponential behavior more plausible, i.e. there is a mixture of shorter lived single and longer-lived multiple receptor-ligand interactions in the data) which would make the catch bond finding here more qualitative than quantitative, i.e. it cannot be excluded that this behavior is caused by multiple or clustering of SpsD molecules. This option would weaken some of the claims made about the catch bond, but the fundamental novelty of the finding still clearly holds.
- additional experiments in which the tethering geometry of the fibrinogen ligand is more tightly controlled through site-specific attachment and/or a negative force-clamp control in which the cantilever is modified with a non-SpsD-interacting control protein (e.g. fibronectin, BSA or similar) which could elucidate if there is a non-specific interaction or pulling geometry which might explain the double exponential. This is the most involved option.

Regardless, I believe the authors will be able to resolve this quickly.

Reviewer #3 (Remarks to the Author):

In the revised manuscript, the authors have addressed all concerns of the reviewers and revised the manuscript accordingly. The paper is now ready for publication. Congratulations to the authors for this excellent and exciting paper.

Reviewer #2 (Remarks to the Author):

The revisions given by the authors leave no major concerns from my side. The manuscript is ready for publication if the very minor comments below are addressed. Again, congratulations to the authors. An assessment of R3 comments can also be found below. We thank the reviewer for the positive feedback and his approval for publication of our manuscript. We have addressed point-by-point the concerns below.

Minor comments:

Line 135-136: Is the affinity of SpsD known? The affinity of DLL adhesins is in the micromolar range, so not exceptionally high. The koff rates here do not relate to the bulk affinity of the system as they are measured under force. In general, single-molecule mechanics and bulk binding kinetics are not correlated (e.g. systems of much higher affinity have lower mechanical stabilities and a much faster koff0, notably Streptavidin:Biotin). Please rephrase. The dissociation constant for the interaction between SpsD and Fg was determined by SPR experiments in a previous study by Pietrocola *et al.* (PLoS ONE 8, e66901 (2013)) and found to be $0.360 \pm 0.032 \mu\text{M}$. We totally agree that comparison between bulk binding kinetics and results presented in this manuscript is hazardous as the later are obtained under external force. Here the reviewer is missing that we do not compare bulk experiments with single molecule data, but similar systems, i.e. SpsD-Fg with SdrG-Fg as requested by the reviewer in the previous revision. We have clarified the text accordingly in lines 135-138.

Figure S1 should be more explicitly labeled (exact surface/cantilever bound molecules/exact cell conditions). Figure S1 (and figure S2) just contains extra cells to support the information shown in Figure 1, where everything is detailed in both the text and methods. For SCFS and SMFS (Fig. S1 and S2), the gold surface and tips were functionalized through EDC/NHS chemistry, as mentioned and detailed in the Methods sections. All measurements were performed at room temperature, in PBS buffer and at a retraction velocity of $1,000 \text{ nm}\cdot\text{s}^{-1}$, as specified in the legends. We have added in insets the spring constant of the cantilever used for the corresponding cells.

Figure S6, please provide a zoom in on the force traces, the constant force plateau is barely visible to judge the stability of the force clamp (i.e. please explicitly show plots of the constant force region in the range of the respective clamping force). Currently the features in the trace are not recognizable. To address this concern, we have added a panel B in Fig. S6 (also shown in fig. R1 below) a zoom in on the clamping part – we kept the force scale constant (100 pN around the central value) and the time scale constant (a window of 0.6 s, as labelled in the x-axis) for each curve for comparison.

Figure R1. Force vs time curves for each clamping force in the clamping region, over a constant period of time of 0.6s. This shows that the clamping was stable.

Please also show the extension vs. time traces to highlight the speed of the feedback on the cantilever position, as this was a major point of discussion in the initial review. As shown in figure R2 below, extension vs time curves did not show any significant deviation or oscillation while clamping the force. As above, for easier comparison, we have kept the tip position scale constant (20 nm) and the time scale constant (a window of 0.6 s, as labelled in the x-axis).

Figure R2. Extension vs time curves for each clamping force in the clamping region, over a constant period of time of 0.6s. This shows that the clamping was stable both in terms of force (as shown in Fig R1) and extension.

The explicitly shown fit of the e-WLC model to force-extension data with explicit parameters fitted is not included yet, it must be added it in the final version. We don't believe that showing a fit of the e-WLC on a single curve in the main manuscript provide valuable information for the story. In the

manuscript, we have already provided the distribution of the contour lengths. Shown below in figure R3, three representative force curves, overlapped with the eWLC fit and the fitting parameters.

Figure R3. Force vs tip position curves overlapped with the eWLC fit of the adhesion peak.

It would be advisable to add all data shown in response to R3 to the supplement. What is the x-axis in and the unit of the heatmap shown in Fig. R1? These plots are somewhat difficult to decipher. The raw data of the force clamps should be published alongside the manuscript so that in the future the discussion with R3 might be resolved through different model fits.

We thank the reviewer for the advice. R3 has been satisfied with the answer and accepted the manuscript at this stage. We consider this is a technical issue that do not change the main message of the journal, but including this information would turn the paper into a more difficult a complex story in which our main achievement that is the existance of a catch bond in gram positive bacteria would be diluted.

We thank the reviewer for the advice. R3 has been satisfied with the answer and accepted the manuscript at this stage. We consider this is a technical issue that does not change the main message, including this information would turn the paper into a more difficult and omplex story in which our main achievement that is the existance of a catch bond in gram positive bacteria would be diluted. We would like to highlight that the reviewers comments are available along with the manuscript and supporting information (SI) in the Nature Communication journal. As a consequence, we feel it is not needed to include all the figures devoted to answer the reviewer in the supporting materials, especially if they do not directly support our findings. Nonetheless, we have now included in the SI file the figure showing that a bi-exponential decay cannot fit our data properly (new Fig. S7), and thus do not explain our claims of a catch bond occurring at extremely high forces $\sim 1,100$ pN and likely involving an unconventional model where propagation and direction of the applied force play central roles.

In regard to the major concerns raised by R3...

As reviewer 3 is fully satisfied there is no point to go back to such discussion.

Reviewer #3 (Remarks to the Author):

In the revised manuscript, the authors have addressed all concerns of the reviewers and revised the manuscript accordingly. The paper is now ready for publication. Congratulations to the authors for this excellent and exciting paper. We thank the reviewer for the positive feedback and his approval for publication of our manuscript.